# (Sparse) Attention to the Details:
# Preserving Spectral Fidelity in ML-based Weather Forecasting Models

**Maksim Zhdanov**[1 2]    **Ana Lucic**[2]    **Max Welling**[1 2 3]    **Jan-Willem van de Meent**[1 2]

## Abstract

We introduce MOSAIC, a probabilistic weather forecasting model that addresses three failure modes of spectral degradation in ML-based weather prediction: spectral damping (statistical), high-frequency aliasing (architectural), and residual high-frequency leakage (parametric). MOSAIC generates ensemble members through learned functional perturbations and operates on native-resolution grids via mesh-aligned block-sparse attention, a hardware-aligned mechanism that captures long-range dependencies at linear cost by sharing keys and values across spatially adjacent queries. At 1.5° resolution with 214M parameters, MOSAIC matches or outperforms models trained on 6× finer resolution on key variables and achieves state-of-the-art results among 1.5° models, producing well-calibrated ensembles whose individual members exhibit near-perfect spectral alignment across all resolved frequencies. A 24-member, 10-day forecast takes under 12 s on a single H100 GPU. Code is available at github.com/maxxxzdn/mosaic.

## 1. Introduction

Accurate weather forecasts save lives and enable timely decisions during extreme events. Phenomena such as frontal zones and tropical cyclones cause catastrophic damage and span relatively short distances, requiring models that faithfully resolve fine spatial scales. Numerical weather prediction (NWP) systems achieve this by integrating the equations of fluid dynamics, thermodynamics, and radiative transfer, but at a computational cost that scales cubically with grid resolution.

[1]AMLab [2]University of Amsterdam [3]CuspAI. Correspondence to: Maksim Zhdanov <m.zhdanov@uva.nl>.

*Proceedings of the 43rd International Conference on Machine Learning*, Seoul, South Korea. PMLR 306, 2026. Copyright 2026 by the author(s).

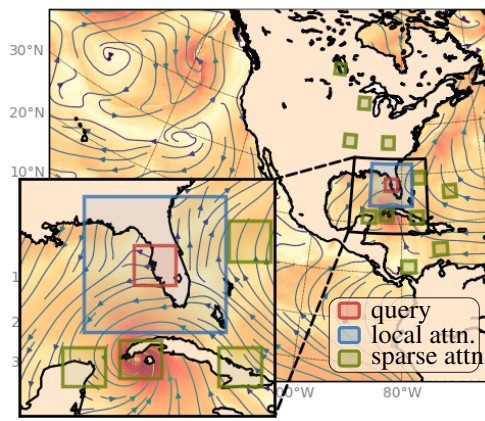

*Figure 1.* Block-sparse attention for weather forecasting. Spatially close query tokens (red block over Tampa Bay) collectively attend to both local key-value pairs (blue block over Florida) and dynamically selected, spatially distributed ones (green blocks). Sparse attention enables capturing long-range dependencies in high-resolution weather data, critical for extreme events such as hurricane formation (note the eye visible in the inset).

ML-based weather prediction models (MLWPs; Bi et al. 2023; Lam et al. 2023; Bodnar et al. 2025) have emerged as efficient surrogates, generating 10-day forecasts in under 60 seconds on a single GPU, achieving a 1000-10000× speedup over NWP (Buizza et al., 2018; Bauer et al., 2020). However, these models fail to faithfully reproduce the spectral signature of weather data at fine scales (Bonavita, 2024), which happens for two main reasons. First, most MLWPs are deterministic and are trained to approximate the conditional mean over future states, which is inherently smoother than any individual realization. Probabilistic MLWPs (Price et al., 2023; Lang et al., 2024; Alet et al., 2025) address this limitation by producing ensemble members rather than a single mean prediction. Each member represents a plausible realization that can exhibit the sharp, fine-scale structures with spectral properties substantially closer to ground truth than their deterministic counterparts.

Second, most MLWPs use compressive encoding in their model architecture: they project high-resolution weather data onto a coarser latent mesh before processing, with a spatial coarsening factor that typically far exceeds any compensating increase in feature channels. This bottle-

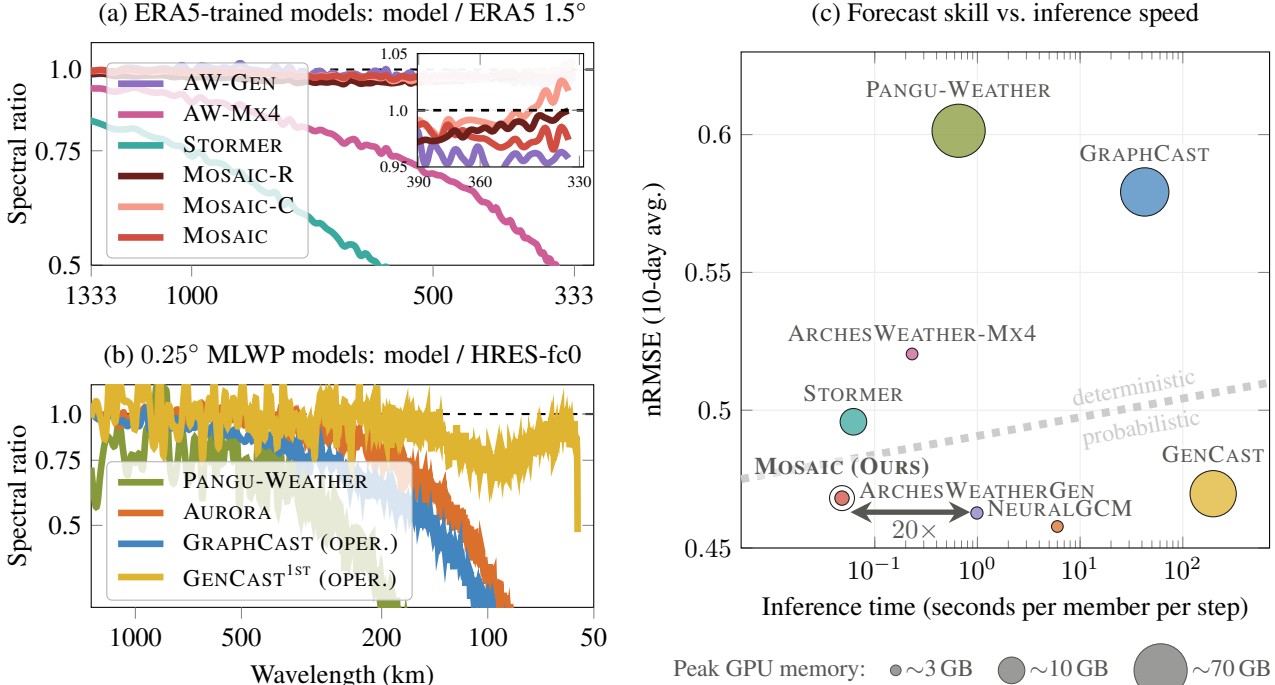

*Figure 2.* Spectral analysis and efficiency of MLWP models. **(a)** Spectral power ratios (model / ERA5) of 10-meter wind speed at $1.5°$ resolution, aggregated over 720 initial conditions throughout the 2020 test year (00:00 and 12:00 UTC) at 24 h lead time, with 16 ensemble members for probabilistic models. **(b)** Same quantity at $0.25°$ (HRES-fc0) from a single 6 h forecast. Probabilistic models (MOSAIC, ARCHESGEN, GENCAST) track the reference closely, while deterministic models (STORMER, ARCHESWEATHER-MX4, GRAPHCAST, AURORA, PANGU-WEATHER) suppress fine-scale energy. MOSAIC-C (a compressed variant entering at a coarser latent grid) and MOSAIC-R (a residual variant predicting $x_{t+1} - x_t$ rather than $x_{t+1}$ directly) instead exhibit the opposite signature, with energy rising back toward 1.0 at the highest wavenumbers, driven by two distinct failure modes: high-frequency aliasing for MOSAIC-C, and residual high-frequency leakage for MOSAIC-R. **(c)** Forecast skill vs. inference speed on a single H100 GPU; marker size indicates peak GPU memory. nRMSE is computed at 240 h lead time at $1.5°$ resolution over a set of key variables (see Appendix C.9 for details).

neck has been shown to cause irreversible information loss in both vision (Wang et al., 2025) and weather models (Nguyen et al., 2023). Beyond information loss, compressive encoding introduces a second failure mode: when the latent mesh is too coarse to represent fine-scale atmospheric features, nonlinear operations alias their energy into lower-wavenumber modes, which re-emerge as spurious high-frequency content upon decoding (McCabe et al., 2023; Fanaskov & Oseledets, 2022; Michaeli & Soudry, 2025). Unlike deterministic smoothing, which suppresses fine-scale spectral energy, this architectural failure mode amplifies it. This signature is visible in GEN-CAST (Fig. 2a), which, despite avoiding deterministic smoothing, exhibits a non-monotone bump in spectral energy near the Nyquist limit.

Finally, the choice of output parametrization introduces a third, parametric failure mode. Predicting tendencies $r(x_t)$ and recovering the next state as $x_{t+1} = x_t + r(x_t)$ (Lam et al., 2023; Bi et al., 2023) leads to high-frequency error accumulation, as any error persists in $x_t$ with high-frequency errors in $r$ added on top, thus compounding over long autoregressive rollouts (Bonev et al., 2025). We re-

fer to the failure mode as residual high-frequency leakage. Predicting the next state directly avoids the carry-over.

In this work, we introduce MOSAIC: a probabilistic weather forecasting model designed to address all three failure modes of spectral degradation, namely spectral damping (statistical), high-frequency aliasing (architectural), and residual high-frequency leakage (parametric). First, following (Alet et al., 2025), we use learned functional perturbations to incorporate uncertainty, producing ensemble forecasts whose individual members preserve realistic spectral variability. Second, we propose block-sparse attention: a hardware-aligned formulation of native sparse attention (Yuan et al., 2025) that exploits the intrinsic locality of physical data by sharing keys and values across spatially adjacent queries, computing block-to-block interactions. Each block dynamically selects which regions of the grid to attend to, capturing arbitrarily long-range dependencies at linear cost. To ensure efficient memory access, we process data on the HEALPix mesh (Gorski et al., 1999), which offers contiguous storage and therefore allows us to operate on blocks. Third, MOSAIC predicts the next weather state directly rather than the residual, avoiding

residual high-frequency leakage in long rollouts. Together, these design choices make MOSAIC directly applicable to high-resolution grids, with spatial interactions captured at native resolution before any coarsening occurs.

The main contributions of this work are:

- We propose mesh-aligned block-sparse attention, a sparse attention mechanism that captures long-range dependencies at linear cost by jointly selecting key-value blocks for spatially adjacent queries grouped along a physical mesh. This enables weather models to compute spatial interactions at native resolution.

- We introduce MOSAIC, a probabilistic weather forecasting model that uses block-sparse attention, learned functional perturbations, and direct next-state prediction to address all three failure modes of spectral degradation in MLWPs: spectral damping (statistical), high-frequency aliasing (architectural), and residual high-frequency leakage (parametric).

- We demonstrate that MOSAIC at 1.5° resolution matches or outperforms models trained on 6× finer resolution on headline upper-air variables, produces well-calibrated ensembles with near-perfect spectral alignment, and generates a 24-member, 10-day forecast in under 12 s on a single H100 GPU.

## 2. Related Work

### 2.1. Effective Resolution in Weather Forecasting

Abdalla et al. (2013) define the effective resolution of a weather prediction model as the smallest spatial scale it fully resolves – the wavelength at which the model's power spectrum starts to deviate from the ground truth. Multiple studies (Bonavita, 2024; Husain et al., 2025) show that MLWPs fail to reproduce realistic spectra and therefore exhibit low effective resolution, struggling to resolve sharp phenomena such as frontal zones and tropical cyclones that span 50-80 km, despite being trained on data at 28 km spatial resolution. Gupta et al. (2025) find systematic underestimation of spectral power at mesoscales (10-100 km) across multiple MLWPs. Li et al. (2025) demonstrate that Pangu (Bi et al., 2023) underestimates kinetic energy at wavelengths below 1000 km and fails to replicate the characteristic $-\frac{5}{3}$ spectral slope of physics-based models.

Several strategies address this spectral degradation. Subich et al. (2025) modify the training objective to penalize spectral discrepancy directly, improving GraphCast's effective resolution from 1,250 to 160 km. Similarly, Bonev et al. (2025) optimize in the spectral domain, achieving realistic spectra at subseasonal lead times. Hybrid approaches (Kochkov et al., 2023; Husain et al., 2025) combine ML with physics-based solvers, leveraging the numerical backbone for fine-scale structure. Post-hoc meth-

ods (Lippe et al., 2023; Oommen et al., 2024) condition diffusion models on smooth predictions to recover high-frequency content. Most directly related to our work, Baño-Medina et al. (2025); Nordhagen et al. (2025) operate on the original high-resolution mesh with message-passing neural networks and observe significantly better spectral correspondence. We follow the same principle, but replace the fixed graphs with sparse attention, which dynamically determines interactions based on the current weather state.

### 2.2. Ultra-scale Grid Processing

Avoiding compression shifts the bottleneck to efficiently processing the original ultra-scale data. Standard attention (Vaswani et al., 2017) scales quadratically with sequence length; FlashAttention (Dao et al., 2022) reduces the memory cost to linear, but still has quadratic complexity. Linear attention (Yang et al., 2024; 2023) achieves linear cost by replacing the softmax with a kernel-based approximation that maintains a fixed-size state, but sacrifices the input-dependent selectivity that makes standard attention expressive. Native Sparse Attention (NSA; Yuan et al. 2025) resolves this trade-off by computing the dot product over a dynamically selected subset of key-value pairs per query, retaining softmax expressivity at linear cost.

We adapt sparse attention for high-resolution weather grids but further exploit spatial locality: rather than selecting key-value blocks independently per query, we group spatially proximate queries into blocks that jointly select which regions to attend to. Concurrent and independent of our work, Gu et al. (2026) and Meituan (2025) develop closely related block-shared sparse attention for video diffusion, and Wang et al. (2026) for long-context language models; these methods target sequence or regular-grid data, where blocks of contiguous indices already correspond to spatial or temporal neighborhoods. We instead target physical data on the sphere, where this property does not hold on the standard latitude-longitude grid; we recover it by operating on the HEALPix mesh with NESTED indexing, so that contiguous-index blocks coincide with spatial neighborhoods on the sphere. This block-level sparsity aligns well with GPU memory access patterns, enabling efficient processing of grids with hundreds of thousands of points without domain decomposition or lossy pooling.

## 3. Theoretical Background

### 3.1. Problem Formulation

We frame weather forecasting as sampling from the conditional distribution $p\left(X^t \mid X^{t-1}, \ldots, X^{t-H}\right)$ over the next atmospheric state $X^t$ given the history of previous states, where each state consists of both surface- and pressure-level atmospheric variables on a latitude-longitude grid. A

$T$-step ensemble forecast is generated by sampling autoregressively:

$$X^t \sim p\left(X^t \mid X^{t-1}, \ldots, X^{t-H}\right), \quad t = 1, \ldots, T. \quad (1)$$

Drawing multiple trajectories from this process yields an ensemble that quantifies forecast uncertainty.

### 3.2. Hierarchical Sphere Tessellation

HEALPix (Gorski et al., 1999) is a hierarchical, equal-area tessellation of a sphere into four-sided polygons (pixels). The sphere is partitioned into a tree-like hierarchy starting from 12 base pixels (4 around each pole, 4 around the equator), with each pixel recursively subdivided into four children (Fig. 10 in App. C.3). At a given resolution, each pixel covers an identical surface area, ensuring that signal sampling and noise integration are geographically unbiased, unlike traditional latitude-longitude grids, which over-sample near the poles.

The size parameter $N_{\text{side}}$ defines a HEALPix mesh and determines both the total number of pixels $12N_{\text{side}}^2$ and the approximate angular resolution. Under the NESTED ordering scheme, pixels are organized in memory such that spatially close regions occupy consecutive indices: a pixel with index $p$ at resolution $N_{\text{side}}$ subdivides into four children with consecutive indices $4p, 4p+1, 4p+2, 4p+3$ at resolution $2N_{\text{side}}$. This contiguous, hierarchical layout makes HEALPix particularly suited for block-based computation: loading a spatially local block of pixels requires a single coalesced memory read rather than the scattered accesses needed on a latitude-longitude grid. Several ML-based weather models already exploit this structure (Ramavajjala, 2024; Linander et al., 2025; Karlbauer et al., 2023). The choice of entry resolution has spectral consequences: halving $N_{\text{side}}$ halves the Nyquist wavenumber of the latent grid, so features at finer scales cannot be faithfully represented and their energy may re-emerge as aliased high-frequency content upon decoding (Appendix C.11).

### 3.3. Native Sparse Attention

Native Sparse Attention (NSA; Yuan et al. (2025)) addresses the quadratic complexity of standard self-attention by restricting each query's interactions to a dynamically determined subset of keys and values. The computation is organized into three branches – a *compression* branch that attends to coarse-grained block representations and provides scores that guide sparsification, a *selection* branch that attends at full resolution to the top-$n$ blocks chosen by those scores, and a *local* branch that applies standard attention within a sliding window. Their outputs $\mathbf{o}_i^{CG}, \mathbf{o}_i^{FG}, \mathbf{o}_i^L$ are combined via learned gating

$$\mathbf{o}_i = g^{CG}(\mathbf{x}_i) \cdot \mathbf{o}_i^{CG} + g^{FG}(\mathbf{x}_i) \cdot \mathbf{o}_i^{FG} + g^L(\mathbf{x}_i) \cdot \mathbf{o}_i^L, \quad (2)$$

where $g^{CG}, g^{FG}, g^L$ are learnable linear functions. Per-branch equations are given in Appendix C.6.

## 4. MOSAIC

Tobler's first law of geography states that "nearby things are more related than distant things" (Tobler, 1970). This principle motivates two important design choices in MOSAIC: (1) representing data on the HEALPix mesh, where spatially close points occupy contiguous memory, and (2) grouping spatially adjacent tokens into blocks that jointly determine which regions of the globe to attend to.

### 4.1. Data Representation on HEALPix

Weather forecast data is traditionally stored on latitude-longitude grids, where points are ordered row-wise along parallels. This layout places spatially close points far apart in index space, requiring scattered memory accesses to load a block of neighboring pixels. We instead operate on the HEALPix mesh, which guarantees that spatial neighbors occupy contiguous memory locations, enabling coalesced GPU reads and hardware-aligned block computation.

**Interpolating Between Grids** To transfer data from the latitude-longitude grid to the HEALPix mesh, we employ the cross-attention interpolation scheme of (Wessels et al., 2025). For each target point $i$ on the HEALPix mesh and neighboring source point $j$ on the latitude-longitude grid, queries are computed from the relative position $\mathbf{p}_{ij}$, while keys and values are derived from source features:

$$\mathbf{q}_{ij} = W_q \frac{\mathbf{p}_{ij}}{\|\mathbf{p}_{ij}\|}, \quad \mathbf{k}_j = W_k \mathbf{x}_j, \quad \mathbf{v}_j = W_v \mathbf{x}_j \quad (3)$$

The interpolated output at target position $i$ is:

$$\mathbf{o}_i = \sum_{j \in \mathcal{N}_i} \text{softmax}_j \left(\mathbf{q}_{ij}^\top \mathbf{k}_j / \sqrt{d}\right) \mathbf{v}_j \quad (4)$$

where $\mathcal{N}_i$ denotes the set of source grid points neighboring $i$.

### 4.2. Block-Sparse Attention

BSA retains the three-branch structure of NSA – compression, selection, and local – combined via learned gating (Eq. 2). The key difference is that sparsity operates at the block level: rather than each query token independently selecting which key blocks to attend to, tokens within a query block jointly select key blocks. This reduces compression from token-to-block to block-to-block interactions and amortizes the selection cost across all tokens in a block.

**Compression** We partition $N$ tokens into $m$ non-overlapping blocks $\{B_1, \ldots, B_m\}$ and compute coarse-grained representations of queries, keys, and values via a

| a) **Interpolation** 
 lat-lon → HEALPix | b) **Compressed attention** 
 between blocks of size 4 | c) **Selective attention** 
 to top-3 blocks | d) **Local attention** 
 with block size 16 |
|---|---|---|---|
| 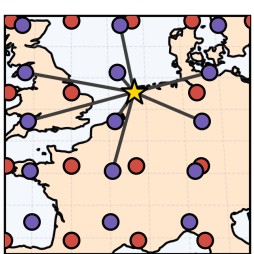 | 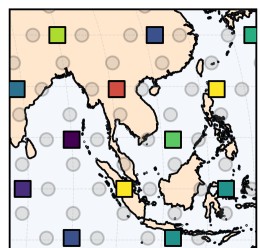 | 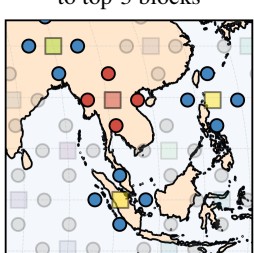 | 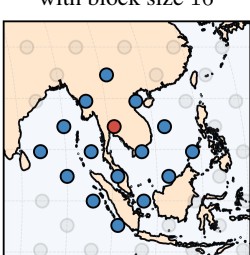 |

*Figure 3.* Block-sparse attention for weather forecasting. **(a)** Weather data is interpolated from a latitude-longitude grid (red) to the HEALPix mesh (purple) via cross-attention. **(b–d)** The three branches of block-sparse attention, illustrated for a single query block (red): **(b)** Compression computes attention between coarse-grained block representations (squares; color indicates attention score). **(c)** Selection attends at full resolution to the top-n key blocks. **(d)** Local attention captures fine-grained interactions within local blocks.

function $\varphi$ (mean pooling in our experiments):

$$\bar{\mathbf{q}}_i = \varphi(\mathbf{Q}_i), \quad \bar{\mathbf{k}}_j = \varphi(\mathbf{K}_j), \quad \bar{\mathbf{v}}_j = \varphi(\mathbf{V}_j). \quad (5)$$

Attention is computed at the block level between query block $i$ and key block $j$:

$$\bar{a}_{ij} = \frac{\exp\left(\bar{\mathbf{q}}_i^\top \bar{\mathbf{k}}_j / \sqrt{d_k}\right)}{\sum_{l=1}^m \exp\left(\bar{\mathbf{q}}_i^\top \bar{\mathbf{k}}_l / \sqrt{d_k}\right)} \quad (6)$$

The coarse-grained output is then distributed to all tokens within each query block:

$$\bar{\mathbf{o}}_i^{CG} = \sum_{j=1}^m \bar{a}_{ij} \bar{\mathbf{v}}_j, \qquad \mathbf{o}_l^{CG} = \bar{\mathbf{o}}_i^{CG} \quad \forall l \in B_i. \quad (7)$$

Coarse-grained attention captures broad spatial patterns in a computationally feasible manner. For instance, a block over the Netherlands might attend strongly to blocks over the North Atlantic or the Arctic, identifying synoptic-scale influences before the selection branch zooms in on details.

**Selection** For each query block $i$, the top-$n$ key blocks with the highest coarse-grained attention scores are selected, $\bar{\mathcal{S}}_i = \text{top-}n\left(\bar{a}_{i,:}\right)$. Fine-grained attention is then computed between all tokens in query block $i$ and all tokens in the selected key blocks, capturing long-range dependencies:

$$\mathbf{o}_l^{FG} = \sum_{j \in \bar{\mathcal{S}}_i} \sum_{t \in B_j} \frac{\exp\left(\mathbf{q}_l^\top \mathbf{k}_t / \sqrt{d_k}\right)}{Z_l} \mathbf{v}_t \quad \forall l \in B_i \quad (8)$$

where $Z_l = \sum_{j \in \bar{\mathcal{S}}_i} \sum_{t \in B_j} \exp\left(\mathbf{q}_l^\top \mathbf{k}_t / \sqrt{d_k}\right)$ is the normalizing constant. The selection branch resolves fine-scale structure within the chosen regions, for example, by capturing how a specific frontal zone over the Atlantic influences local conditions in Western Europe.

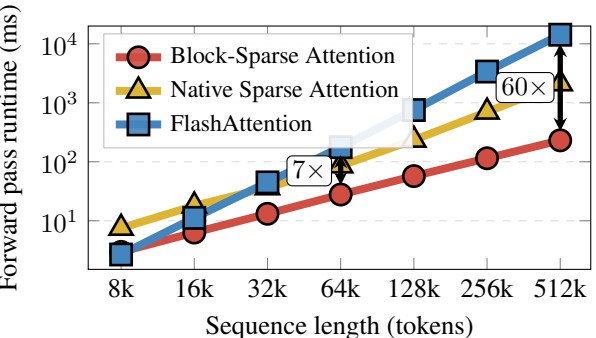

*Figure 4.* Forward pass runtime vs. sequence length for Block-Sparse Attention, NSA, and full FlashAttention; measured on NVIDIA RTX A4500. See Appendix C.3 for details.

**Local Attention** To capture fine-grained local interactions, we compute attention within large blocks of pixels independently. Unlike NSA's sliding window, which would require handling irregular boundaries on the sphere, block attention aligns naturally with the HEALPix structure and can be computed in parallel. The local branch resolves fine-grained structure within each region, such as temperature gradients across a coastline or wind patterns within a valley, freeing other branches to focus on long-range interactions.

**Computational cost** Let $b$ denote the block size yielding $\frac{N}{b}$ blocks in total. The combined cost of block-sparse attention across branches is: $\mathcal{O}\left(N^2/b^2 + Nnb + Nb\right)$. The compression branch (CG) computes all pairwise interactions between coarse-grained blocks, resulting in $\mathcal{O}(N^2/b^2)$ cost. For block sizes $b \geq 128$, this cost does not form a bottleneck. The selection branch (FG) computes, for each token, attention over $n$ selected blocks of size $b$, yielding $\mathcal{O}(Nnb)$ cost, which is linear in $N$ as $n$ and $b$ are fixed hyperparameters. Local attention is $\mathcal{O}(Nb)$, since each token only attends to tokens within the same block. Together, these costs make block-sparse attention scalable to weather grids with hundreds of thousands of tokens. In practice,

BSA scales near-linearly with sequence length, achieving up to $61.8\times$ speedup over dense attention and $9.4\times$ over NSA (Fig. 4). We implement BSA in Triton (Tillet et al., 2019) following the memory-efficient approach of FlashAttention (Dao et al., 2022); see Appendix C.3 for details.

### 4.3. Model Architecture

Weather dynamics spans a wide range of spatial scales, from planetary waves spanning thousands of kilometers to mesoscale convective systems tens of kilometers across. To capture this multi-scale structure, we adopt a U-Net architecture that processes data at progressively coarser resolutions in the encoder path, then refines predictions back to the original resolution in the decoder path. Each resolution level consists of block-sparse attention layers that capture interactions at that scale. Skip connections between corresponding encoder and decoder levels preserve fine-grained information that might otherwise be lost during coarsening. Crucially, the first encoder stage computes spatial interactions via block-sparse attention at native resolution before any coarsening occurs. This distinguishes MOSAIC from compression-based MLWPs, which first reduce spatial resolution (e.g., via patchification) and then compute interactions on the compressed representation, creating an information bottleneck at the finest scales. By acting on the native grid before coarsening, MOSAIC avoids the aliasing that arises when nonlinear operations are applied below the Nyquist limit of the native resolution (Appendix C.11).

**Coarsening and refinement** The encoder path progressively coarsens the HEALPix mesh: at each step, the four children at resolution $2N_{\text{side}}$ are pooled into their parent at $N_{\text{side}}$ via a learnable position-aware projection. The decoder path applies the transpose, with encoder features added through skip connections. A bottleneck stage applies additional transformer blocks at the coarsest resolution. Full equations are in App. C.3.

**Overall architecture** Dynamic variables (e.g., 2-meter temperature, 10-meter wind), static variables (e.g., land-sea mask, soil type), and sinusoidal time encodings (Bodnar et al., 2025) are projected to the hidden dimension via a two-layer MLP and interpolated to the HEALPix mesh (Eq. 4). The data then passes through the U-Net, where each resolution level applies a sequence of pre-norm transformer blocks (Touvron et al., 2023) with block-sparse attention and SwiGLU (Shazeer, 2020) feed-forward networks. We use grouped query attention (GQA; (Ainslie et al., 2023)) and 2D rotary positional embeddings (RoPE; (Heo et al., 2024)) on queries and keys, with the head dimension split equally between longitude and latitude. After the decoder, features are interpolated back to the latitude-longitude grid and projected to predict the next state $x_{t+1}$

directly rather than the residual tendency $x_{t+1} - x_t$, avoiding residual high-frequency leakage (Bonev et al., 2025).

### 4.4. Uncertainty Quantification

Block-sparse attention addresses the architectural source of spectral degradation by capturing spatial interactions at native resolution before any coarsening, avoiding the compress-first bottleneck. To address the statistical source, MOSAIC produces ensemble forecasts via learned functional perturbations (Alet et al., 2025), a form of Bayesian neural networks (Blundell et al., 2015). The key idea is to inject noise into the parameters of the model, resulting in a single weight perturbation affecting the entire forecast in a globally consistent way. Alet et al. (2025) condition parameters of layer normalization layers on the noise vector $\mathbf{z}$; we instead inject $\mathbf{z}$ as an additive bias inside the gate of every SwiGLU layer (cSwiGLU), which we found to work best (App. C.3).

**Training with CRPS** The probabilistic model is trained by optimizing the latitude-weighted, variable-weighted unbiased (fair) Continuous Ranked Probability Score (Alet et al., 2025; Zamo & Naveau, 2018), a proper scoring rule that decomposes into a reliability term penalizing deviation from the ground truth and a sharpness term encouraging ensemble spread only where warranted by true uncertainty. Per-channel weights $\alpha_i$ follow Lam et al. (2023), latitude weights satisfy $\omega_h \propto \cos(\phi_h)$, and predictions and targets enter the loss in standardized space. Full estimator and weights are in App. C.5.

## 5. Experiments

We aim to answer the following research questions:

- **RQ1:** How does MOSAIC compare against SOTA MLWPs in terms of forecasting skill?

- **RQ2:** Does MOSAIC preserve spectral fidelity across all resolved frequencies?

- **RQ3:** Are its ensemble forecasts well-calibrated?

- **RQ4:** Does MOSAIC's spectral fidelity translate into skillful forecasts of real-world extreme events?

We evaluate MOSAIC under two benchmarks that differ in training data, step size, and test year: a controlled $1.5°$ comparison against other $1.5°$ models, and a cross-resolution comparison against state-of-the-art $0.25°$ MLWPs. We further assess extreme-event forecasting through a case study of Hurricane Ian. We also conduct an ablation study (Appendix A) where we modify a single aspect of the model relative to a controlled baseline.

**Data** All experiments share the same pretraining data: ERA5 reanalysis from 1979 to 2018 at $1.5°$ spatial reso-

|  | RES. | STEP | S/STEP† | Z500 | T850 | Q700 | U850 | V850 | SP |
|---|---|---|---|---|---|---|---|---|---|
| IFS HRES | 0.1° | 1H | — | 802.75 | 3.654 | 1848.0 | 6.460 | 6.530 | 748.68 |
| Keisler (2022) | 1° | 6H | — | 787.01 | 3.559 | 1659.5 | 6.106 | 6.121 | N/A |
| Stormer ENS (mean) | 1.4° | 24H | 0.06 | 665.88 | 3.001 | 1445.4 | 5.198 | 5.137 | 619.89 |
| ArchesWeather-Mx4 | 1.5° | 24H | 0.06 | 693.56 | 3.117 | 1541.1 | 5.413 | 5.431 | 641.08 |
| NeuralGCM ENS (50) | 1.4° | 12H | 7.42 | **606.84** | 2.756 | 1374.2 | **4.830** | 4.852 | — |
| ArchesWeatherGen (mean) | 1.5° | 24H | 0.99 | 610.36 | **2.755** | 1373.7 | **4.830** | **4.848** | **567.15** |
| Mosaic (mean; Ours) | 1.5° | 24H | **0.05** | 624.08 | 2.778 | **1358.1** | 4.880 | 4.897 | 576.44 |

*Table 1.* Comparison of 1.5° resolution models on RMSE scores for key weather variables with 10-day lead-time against ERA5 data, 2020 test year. Best scores in **bold**, second best underlined. Results for additional variables (T2m, U10m, V10m) are provided in Table 3.
† Per-member-per-step inference time (seconds) on a single NVIDIA H100. See Table 13 for full benchmarking details.

lution (121×240 equiangular grid). The difference comes down to the finetuning stage:

- **1.5° benchmark.** Following the protocol of Couairon et al. (2024), we finetune on ERA5 2007–2019 to account for distribution shift in the earlier reanalysis period, and test on 2020. Forecasts are evaluated against ERA5 ground truth. MOSAIC is trained with a step size of 24 h to match the baselines.

- **0.25° benchmark.** We finetune on HRES-fc0 analysis from 2016 to 2021 and test on 2022. The evaluation is done by initializing from HRES-fc0, and evaluating against it. MOSAIC is trained with a step size of 6 h as for the majority of the baselines.

**Baselines** For the 1.5° benchmark, we compare against models operating at comparable resolution: Stormer (Nguyen et al., 2024), ArchesWeather-Mx4 (Couairon et al., 2024), ArchesWeatherGen (Couairon et al., 2024), NeuralGCM-ENS (Kochkov et al., 2023). We also include IFS HRES and IFS ENS (ECMWF) as NWP reference points. For the 0.25° benchmark, we compare against deterministic models (GraphCast (Lam et al., 2023), Aurora (Bodnar et al., 2025), Pangu-Weather (Bi et al., 2023)) and probabilistic models (FGN (Alet et al., 2025), GenCast (Price et al., 2023)), as well as IFS-ENS, IFS-HRES, and climatology. Training resources of all baselines are given in Table 7.

**Training details** MOSAIC contains 214M parameters in a U-Net architecture (App. C.3), trained with batch size 2 on 8× NVIDIA H100 GPUs in fp16 with the Muon (Jordan et al., 2024) optimizer. Pretraining takes 250k steps; finetuning runs 60k steps with progressively longer autoregressive rollouts (1, 2, 4, 8, 12 steps) using the pushforward trick (Brandstetter et al., 2022; Bodnar et al., 2025), where only the final unrolled step is backpropagated through. Training-time ensembles have size 2.

**Evaluation** We follow the WeatherBench 2 protocol (Rasp et al., 2023): all forecasts are initialized at 00:00 and 12:00 UTC and regridded to 1.5° before computing metrics. We report latitude-weighted root mean square error (RMSE) of both the first member and the ensemble mean, as well as the fair CRPS (Rasp et al., 2023) of the ensemble (**RQ1**). Metrics are computed at each grid point, then globally area-weighted and averaged for each variable and pressure level separately. To evaluate spectral fidelity, we report spectra of global kinetic energy (**RQ2**). To evaluate ensemble reliability, we report the spread-to-skill ratio – the ratio of ensemble spread to the RMSE of the ensemble mean – where values close to 1 indicate well-calibrated forecasts (**RQ3**). To assess whether spectral fidelity translates into skillful extreme event forecasting, we conduct an ensemble case study of Hurricane Ian using storm-center tracking (Brightband Technologies, 2026) (**RQ4**).

**RQ1: Forecasting skill comparison**

**1.5° benchmark** Table 1 reports 10-day RMSE for all headline variables against ERA5 ground truth. MOSAIC achieves the best Q700 RMSE (1358.1) among all models and the second-best scores on U850 (4.880) and SP (576.44), closely trailing ArchesWeatherGen and Neural-GCM. On Z500, MOSAIC (624.08) narrows the gap to the stronger probabilistic baselines NeuralGCM (606.84) and ArchesWeatherGen (610.36), while substantially outperforming the deterministic models Stormer (665.88) and ArchesWeather-Mx4 (693.56). Notably, MOSAIC achieves this performance at a fraction of the cost, being 20× faster than ArchesWeatherGen and over 150× faster than Neural-GCM. As Fig. 2 (c) shows, MOSAIC matches the forecast skill of the best probabilistic models while being an order of magnitude faster.

**0.25° benchmark** MOSAIC achieves competitive performance against 0.25° SOTA models. Fig. 5 shows RMSE and CRPS results across three headline variables: 2-meter temperature (2m temp), geopotential at 500 hPa (Z500), and 10-meter U-component of wind (U10). At 10-day lead time, MOSAIC achieves strong RMSE performance: on Z500, MOSAIC outperforms IFS-ENS (610 vs 622 $m^2/s^2$),

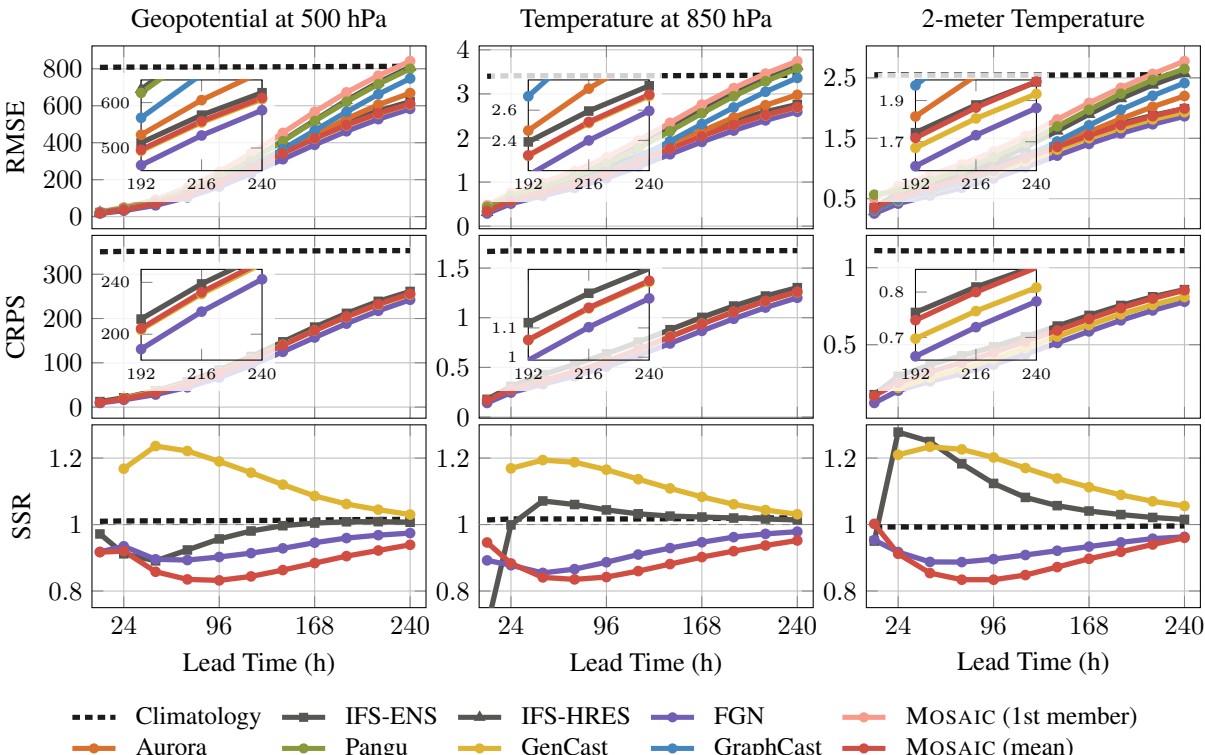

*Figure 5.* Forecast skill evaluated against IFS HRES Analysis following the WeatherBench 2 protocol (Rasp et al., 2023) on 2022 test year. Top row: RMSE; middle row: CRPS; bottom row: SSR.

Aurora (610 vs 668), Pangu (610 vs 800), and GraphCast (610 vs 748); on U10, MOSAIC surpasses IFS-ENS (3.35 vs 3.39 m/s) and all deterministic ML baselines; on 2m temperature, MOSAIC matches IFS-ENS (1.99 K) while outperforming Aurora, Pangu, and GraphCast.

For probabilistic metrics, MOSAIC demonstrates competitive CRPS scores. At 240h, MOSAIC achieves Z500 CRPS of 258, close to GenCast (254) and IFS-ENS (261), though FGN achieves the lowest (242). On U10, MOSAIC (1.58) outperforms IFS-ENS (1.61) and approaches Gen-Cast (1.56). These results demonstrate that block-sparse attention operating at native resolution can extract sufficient information from 1.5° data to compete with models trained on 6× finer grids. Tables 4 and 5 provide RMSE at 24 h and 240 h for a broader set of headline variables; RMSE, CRPS, and spread-to-skill curves for the remaining variables (V10, MSL, U850, V850, Q700) are shown in Appendix B.2. Qualitative unrolling trajectories in Appendix B further illustrate MOSAIC's ability to maintain forecast quality over extended lead times.

**RQ2: Spectral fidelity preservation**

Fig. 2(a) compares kinetic energy spectral ratios of 10-meter wind speed for all ERA5-trained models at 1.5°, aggregated over the 2020 test year (720 initial conditions, 16 ensemble members for probabilistic models) at 24 h

lead time. The three failure modes of spectral degradation produce two opposing signatures: spectral damping suppresses fine-scale energy, while high-frequency aliasing and residual high-frequency leakage both amplify it.

Deterministic models exhibit clear suppression at fine scales (Stormer: 0.24; ArchesWeather-Mx4: 0.47 at the highest resolved wavenumber), the expected outcome of training against an ensemble mean. MOSAIC-C, a compressed variant of MOSAIC entering at $N_{\text{side}}{=}32$ rather than $64$ (otherwise identical), shows the opposite: its ratio grows past $1.0$ near the Nyquist limit ($1.02$), the signature of aliasing where energy not representable on the coarse latent grid folds back upon decoding. The divergence appears near the native Nyquist rather than the hard limit of the coarser grid because the learnable interpolation absorbs part of the unrepresentable content across the feature dimension, mitigating but not eliminating the bottleneck (Appendix C.11). MOSAIC-R, a variant predicting the residual $x_{t+1} - x_t$ rather than $x_{t+1}$ directly (otherwise identical to MOSAIC), shows a similar high-wavenumber rise ($1.00$) driven by residual high-frequency leakage rather than aliasing.

MOSAIC avoids all three failure modes by combining native-resolution interactions, probabilistic training, and direct next-state prediction, tracking the reference closely ($0.97$). ARCHESGEN ($0.96$) likewise shows no spec-

**a)** Init: Sep 23, 2022 12:00 UTC 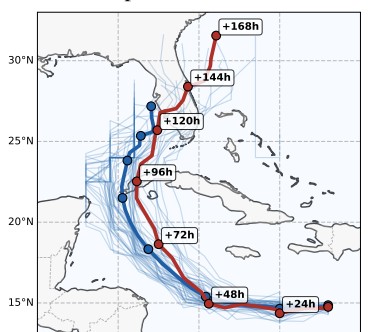 **b)** Init: Sep 25, 2022 12:00 UTC 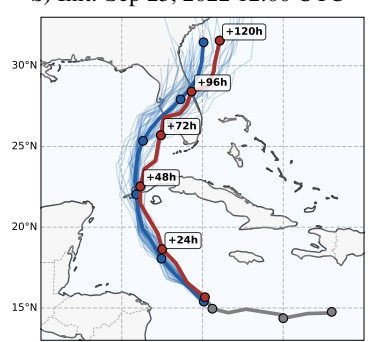 **c)** Init: Sep 27, 2022 12:00 UTC 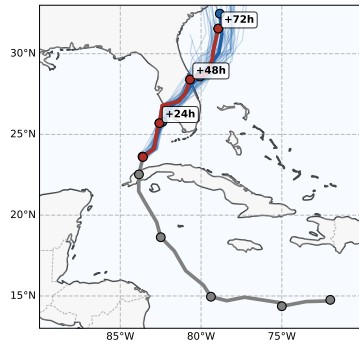

*Figure 6.* Hurricane Ian ensemble track forecasts from three initialization times: Sep 23 (7-day lead), Sep 25 (5-day lead), and Sep 27 (3-day lead), all in 2022. Each panel shows 48 ensemble member tracks (light blue), the ensemble mean (dark blue), and the observed track from HRES-fc0 (black). Markers indicate 24 h intervals. Storm centers are tracked via minimum MSLP guided by IBTrACS best-track positions (Knapp et al., 2010). Wind field evolution is shown in Fig. 8.

tral bump: its deterministic backbone predicts $x_{t+1}$ directly (Couairon et al., 2024), with the diffusion component modeling residuals relative to that backbone rather than to $x_t$. Fig. 2(b) confirms the same patterns at $0.25°$.

### RQ3: Ensemble calibration

Fig. 5 (bottom row) shows the spread-to-skill ratio across lead times. Well-calibrated ensembles should exhibit ratios close to 1, with values below 1 indicating overconfidence and values above 1 indicating underconfidence. MO-SAIC achieves ratios ranging from 0.83 at medium lead times to 0.95 at 240h, closely matching FGN (0.89–0.97), while GenCast shows persistent underconfidence (1.03–1.24). This demonstrates that noise injection in SwiGLU gates produces well-calibrated uncertainty estimates on par with state-of-the-art probabilistic MLWPs.

### RQ4: Extreme event forecasting: Hurricane Ian

To evaluate MOSAIC's ability to forecast extreme weather events, we conduct a case study of Hurricane Ian – a Category 4 hurricane that made landfall near Fort Myers, Florida, on September 28, 2022, with sustained winds of 155 mph. We generate 48-member ensembles from three initialization times (Sep 23, 25, and 27, 2022, all at 12Z) and track each member's storm center by locating the minimum mean sea level pressure (MSLP) within a search radius guided by IBTrACS best-track positions (Knapp et al., 2010), following the methodology of ExtremeWeatherBench (Brightband Technologies, 2026).

Fig. 6 shows the resulting ensemble track forecasts, with wind field evolution shown in Fig. 8. At 7-day lead time (Sep 23 init), the ensemble captures the general northward trajectory through the Caribbean, with multiple members already correctly tracking the recurvature over Cuba and

landfall on the Florida Gulf Coast. At 5-day lead (Sep 25 init), the ensemble spread narrows, and the ensemble mean closely follows the observed track through recurvature and approach to landfall. At 3-day lead (Sep 27 init), the ensemble tightly brackets the observed track, with most members correctly identifying the landfall region. Despite operating at $1.5°$ resolution ($\sim 166$ km), which cannot resolve the inner-core structure of a tropical cyclone, MOSAIC captures the hurricane's evolution. The progressive narrowing of ensemble spread with decreasing lead time demonstrates physically reasonable uncertainty quantification.

## 6. Conclusion

We introduced MOSAIC, a probabilistic weather forecasting model that addresses three failure modes of spectral degradation in MLWPs: spectral damping from deterministic training, high-frequency aliasing from compressive encoding, and residual high-frequency leakage from residual prediction. MOSAIC addresses these through (1) learned functional perturbations, which produce ensemble members with realistic spectral variability, (2) block-sparse attention, which captures spatial interactions at native resolution before any coarsening at linear cost, and (3) direct next-state prediction, which avoids carry-over of high-frequency errors in autoregressive rollouts. MOSAIC matches the forecasting skill of state-of-the-art models on both deterministic and probabilistic metrics, while being an order of magnitude faster. The spectral analysis reveals that individual ensemble members exhibit near-perfect spectral alignment across all resolved frequencies, and a case study of Hurricane Ian shows that this fidelity translates into skillful ensemble track forecasts of real-world extreme events.

## Impact Statement

This paper presents work whose goal is to advance the field of Machine Learning. There are many potential societal consequences of our work, none which we feel must be specifically highlighted here.

## Acknowledgements

MZ acknowledges support from Microsoft Research AI4Science. JWvdM acknowledges support from the European Union Horizon Framework Programme (Grant agreement ID: 101120237). This work used the Dutch national e-infrastructure with the support of the SURF Cooperative using grant no. EINF-16923. Computations were partially performed using the University of Amsterdam - Science Faculty (UvA/FNWI) HPC Facility.

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

# A. Ablation Study

We conduct ablation experiments to validate MOSAIC's key design choices. To make ablations tractable, all variants, including the ablation baseline, are trained under a reduced-scale protocol that differs from the full model (Section C.4) in several ways: training uses ERA5 data from 2007–2018 only (versus 1979–2018), 100k steps with batch size 1 on $4\times$ NVIDIA A6000 GPUs (versus 250k steps with batch size 2 on $8\times$ H100), single-step rollout only (no multi-step finetuning), and no HRES-fc0 finetuning stage. Each ablation modifies exactly one aspect relative to this reduced baseline.

We report the normalized RMSE (nRMSE): for each variable, the latitude-weighted RMSE of the ensemble-mean prediction is divided by the climatological standard deviation of that variable, yielding a dimensionless error on a comparable scale across all quantities. The reported metric is the mean nRMSE across all 82 output variables, evaluated on 6 h forecasts over the 2020 test year following the WeatherBench 2 protocol (Rasp et al., 2023).

*Table 2.* Ablation studies on 6 h forecast quality (mean nRMSE across all 82 output variables). All variants are trained under a reduced-scale protocol (ERA5 2007–2018, 100k steps, $4\times$ A6000, single-step rollout, no HRES-fc0 finetuning). Each ablation modifies exactly one aspect relative to the baseline.

(a) Model scale. Increasing capacity yields steady improvement.

| Variant | Params | nRMSE |
|---|---|---|
| Tiny | 59.8M | 1.0415 |
| Small | 117.9M | 0.1746 |
| Medium | 195.0M | 0.1720 |
| Baseline | 214.3M | 0.1705 |

(b) Sparse attention blocks. More blocks consistently help.

| Sparse blocks | nRMSE |
|---|---|
| 0 / 0 / 0 | 0.1724 |
| 12 / 6 / 2 | 0.1708 |
| 24 / 12 / 4 | 0.1705 |
| 48 / 24 / 8 | 0.1702 |

(c) NSA attention branches. Each branch contributes independently.

| Branches | nRMSE |
|---|---|
| Local only | 0.1724 |
| Local + compression | 0.1720 |
| Local + selection | 0.1723 |
| All three (baseline) | 0.1705 |

(d) Optimizer. Muon outperforms AdamW even with extended budget.

| Optimizer | nRMSE |
|---|---|
| AdamW | 0.1797 |
| AdamW (longer) | 0.1728 |
| Muon (baseline) | 0.1705 |

(e) Training objective. Probabilistic training improves ensemble-mean nRMSE.

| Objective | nRMSE |
|---|---|
| Deterministic (MSE) | 0.1721 |
| Probabilistic (CRPS) | 0.1705 |

(f) History length. Diminishing returns beyond 2 steps.

| History steps | nRMSE |
|---|---|
| 1 | 0.1711 |
| 2 | 0.1699 |
| 4 (baseline) | 0.1705 |

(g) Spatial compression. Avoiding compression preserves forecast quality.

| Entry $N_{\text{side}}$ | nRMSE |
|---|---|
| 32 (compressed) | 0.1771 |
| 64 (baseline) | 0.1705 |

(h) Rotary position embeddings. RoPE provides a consistent benefit.

| Variant | nRMSE |
|---|---|
| No RoPE | 0.1722 |
| RoPE (baseline) | 0.1705 |

**Summary.** Excluding the undersized Tiny model, all ablation variants fall within a narrow nRMSE band of 0.1699–0.1797, indicating that the architecture is generally robust to individual design changes. We discuss each factor below.

*Model scale (a).* The Tiny configuration (59.8M parameters) diverges to an nRMSE of approximately 1.04, revealing a minimum capacity threshold below which training becomes unstable. Beyond this threshold, scaling from Small (117.9M) to the baseline (214.3M) yields steady gains ($0.1746 \rightarrow 0.1705$).

*Sparse attention blocks (b).* Increasing the number of sparse blocks consistently improves nRMSE ($0.1724 \rightarrow 0.1702$), though with diminishing returns. Even without any sparse blocks, performance remains reasonable (0.1724), suggesting that the dense attention layers carry most of the representational load, with sparse blocks providing refinement.

*NSA branches (c).* The three NSA branches – local, compression, and selection – contribute complementarily. Local attention alone yields 0.1724, and adding compression alone brings a modest improvement (0.1720), while adding selection alone barely helps (0.1723). The weak contribution of selection in isolation is expected: the selection branch relies on compressed representations to produce its routing scores, so removing the gradients from the compression branch degrades the guidance signal to essentially no useful information. Combining all three branches yields 0.1705, confirming that all branches are necessary.

*Optimizer (d).* Muon outperforms AdamW at equal training budget (0.1705 vs. 0.1797). However, the advantage is primarily one of convergence speed: with 50% more training steps, AdamW closes most of the gap, reaching 0.1728, within 1.3% of Muon. This makes Muon attractive for the reduced-budget ablation setting, though the two optimizers may converge to more similar performance given sufficient compute.

*Training objective (e).* Probabilistic (CRPS) training improves ensemble-mean nRMSE over deterministic (MSE) training (0.1705 vs. 0.1721), consistent with the observation that optimizing for calibrated ensembles also benefits point forecast accuracy.

*History length (f).* Two input time steps yield the best 6 h nRMSE (0.1699), slightly outperforming the four-step baseline (0.1705). This suggests diminishing returns from longer context at the single-step forecast horizon; the baseline uses four steps to support autoregressive rollouts during finetuning.

*Spatial compression (g).* Entering the U-Net at $N_{\mathrm{side}}$=32 instead of the baseline $N_{\mathrm{side}}$=64 degrades nRMSE by 3.9% ($0.1705 \rightarrow 0.1771$), confirming that native-resolution processing in the first encoder stage is important for preserving fine-grained spatial information. The corresponding spectral analysis (Fig. 2(a)) shows that this degradation is not a loss of fine-scale detail but the emergence of spurious high-frequency energy near the Nyquist limit – the signature of aliasing discussed in Appendix C.11.

*Rotary position embeddings (h).* RoPE provides a consistent benefit, reducing nRMSE from 0.1722 to 0.1705, supporting its use for encoding spatial relationships on the HEALPix grid.

Figure 7 extends the main-text evaluation (Fig. 5) to the remaining surface and pressure-level variables: 10-meter V-wind, mean sea level pressure, U- and V-wind at 850 hPa, and specific humidity at 700 hPa. Rows 1–2 show RMSE, rows 3–4 show CRPS, and rows 5–6 show the spread-to-skill ratio (values close to 1.0 indicate well-calibrated ensembles). All forecasts are regridded to 1.5° resolution.

## B. Additional Results

### B.1. 1.5° Benchmark

Table 3 extends the 1.5° comparison from Table 1 to additional headline variables (T850, U850, V850, SP).

| | RES. | STEP | T2M | U10M | V10M |
|---|---|---|---|---|---|
| IFS HRES | 0.1° | 1H | – | – | – |
| KEISLER (2022) | 1° | 6H | N/A | N/A | N/A |
| STORMER | 1.4° | 24H | 2.239 | 3.535 | 3.735 |
| NEURALGCM ENS (50) | 1.4° | 12H | N/A | N/A | N/A |
| ARCHESWEATHER-MX4 | 1.5° | 24H | 2.323 | 3.716 | 3.924 |
| ARCHESWEATHERGEN (MEAN) | 1.5° | 24H | 2.049 | 3.306 | 3.485 |
| MOSAIC (MEAN; OURS) | 1.5° | 24H | 2.053 | 3.372 | 3.559 |

*Table 3.* Comparison of coarse-resolution MLWPs on RMSE scores for key weather variables with 240 h lead time against ERA5 data, 2020 test year (continuation of Table 1).

### B.2. 0.25° Benchmark

Tables 4 and 5 report per-variable RMSE at 24 h and 240 h lead times for all models evaluated at 1.5° resolution. Figure 7 extends the 0.25° evaluation from Fig. 5 to the remaining surface and pressure-level variables, showing RMSE, CRPS, and spread-to-skill ratio as a function of lead time.

| | RES. | Z500 | T850 | Q700 | U850 | V850 | T2M | SP | U10M | V10M |
|---|---|---|---|---|---|---|---|---|---|---|
| IFS HRES | 0.1° | 41.23 | 0.634 | 537.3 | 1.126 | 1.149 | 0.541 | 58.70 | 0.802 | 0.832 |
| IFS ENS | 0.2° | 40.91 | 0.630 | 503.1 | 1.102 | 1.121 | 0.589 | 59.44 | 0.784 | 0.806 |
| PANGU | 0.25° | 45.01 | 0.681 | 537.2 | 1.180 | 1.218 | 0.624 | 59.54 | 0.796 | 0.825 |
| GRAPHCAST | 0.25° | 38.32 | 0.570 | 475.3 | 1.021 | 1.048 | 0.476 | 49.36 | 0.692 | 0.719 |
| GENCAST | 0.25° | 39.20 | 0.570 | 480.7 | 1.040 | 1.070 | 0.462 | 50.04 | 0.701 | 0.732 |
| FGN | 0.25° | 32.27 | 0.510 | 443.2 | 0.928 | 0.953 | 0.418 | 42.22 | 0.627 | 0.655 |
| AURORA | 0.25° | 38.12 | 0.566 | 477.9 | 1.003 | 1.031 | 0.474 | 48.10 | 0.681 | 0.710 |
| MOSAIC (OURS) | 1.5° | 36.78 | 0.572 | 482.8 | 1.028 | 1.054 | 0.525 | 47.91 | 0.701 | 0.729 |

*Table 4.* RMSE scores for key weather variables at 24 h lead time. All metrics are evaluated at 1.5° resolution.

| | RES. | Z500 | T850 | Q700 | U850 | V850 | T2M | SP | U10M | V10M |
|---|---|---|---|---|---|---|---|---|---|---|
| IFS HRES | 0.1° | 809.0 | 3.626 | 1847. | 6.417 | 6.500 | 2.576 | 753.3 | 4.513 | 4.767 |
| IFS ENS | 0.2° | 621.9 | 2.761 | 1390. | 4.854 | 4.918 | 1.991 | 574.7 | 3.394 | 3.595 |
| PANGU | 0.25° | 799.8 | 3.573 | 1782. | 6.267 | 6.334 | 2.648 | 742.4 | 4.378 | 4.633 |
| GRAPHCAST | 0.25° | 747.6 | 3.364 | 1601. | 5.953 | 6.032 | 2.415 | 698.9 | 4.195 | 4.436 |
| GENCAST | 0.25° | 607.5 | 2.693 | 1340. | 4.758 | 4.826 | 1.932 | 563.2 | 3.325 | 3.521 |
| FGN | 0.25° | 583.1 | 2.595 | 1302. | 4.636 | 4.710 | 1.864 | 542.7 | 3.247 | 3.440 |
| AURORA | 0.25° | 668.4 | 2.979 | 1435. | 5.060 | 5.051 | 2.200 | 624.9 | 3.542 | 3.718 |
| MOSAIC (OURS) | 1.5° | 610.4 | 2.703 | 1359. | 4.773 | 4.828 | 1.992 | 563.2 | 3.339 | 3.534 |

*Table 5.* RMSE scores for key weather variables at 240 h (10-day) lead time. All metrics are evaluated at 1.5° resolution.

### B.3. Hurricane Ian Case Study

Fig. 8 shows the 10-meter wind speed evolution from a single MOSAIC run initialized on September 23, 2022, 12Z (5 days before Hurricane Ian's Category 4 landfall). Ground truth (HRES-fc0 analysis), three individual ensemble members, and the ensemble mean are shown at four lead times (+24 h, +72 h, +120 h, +168 h). Individual members produce distinct realizations of the wind field, while the ensemble mean is smoother, particularly at longer lead times. The intensification and northward progression of the cyclone are visible in both the ground truth and the ensemble members, demonstrating that MOSAIC captures the large-scale evolution of this extreme event despite operating at 1.5° resolution.

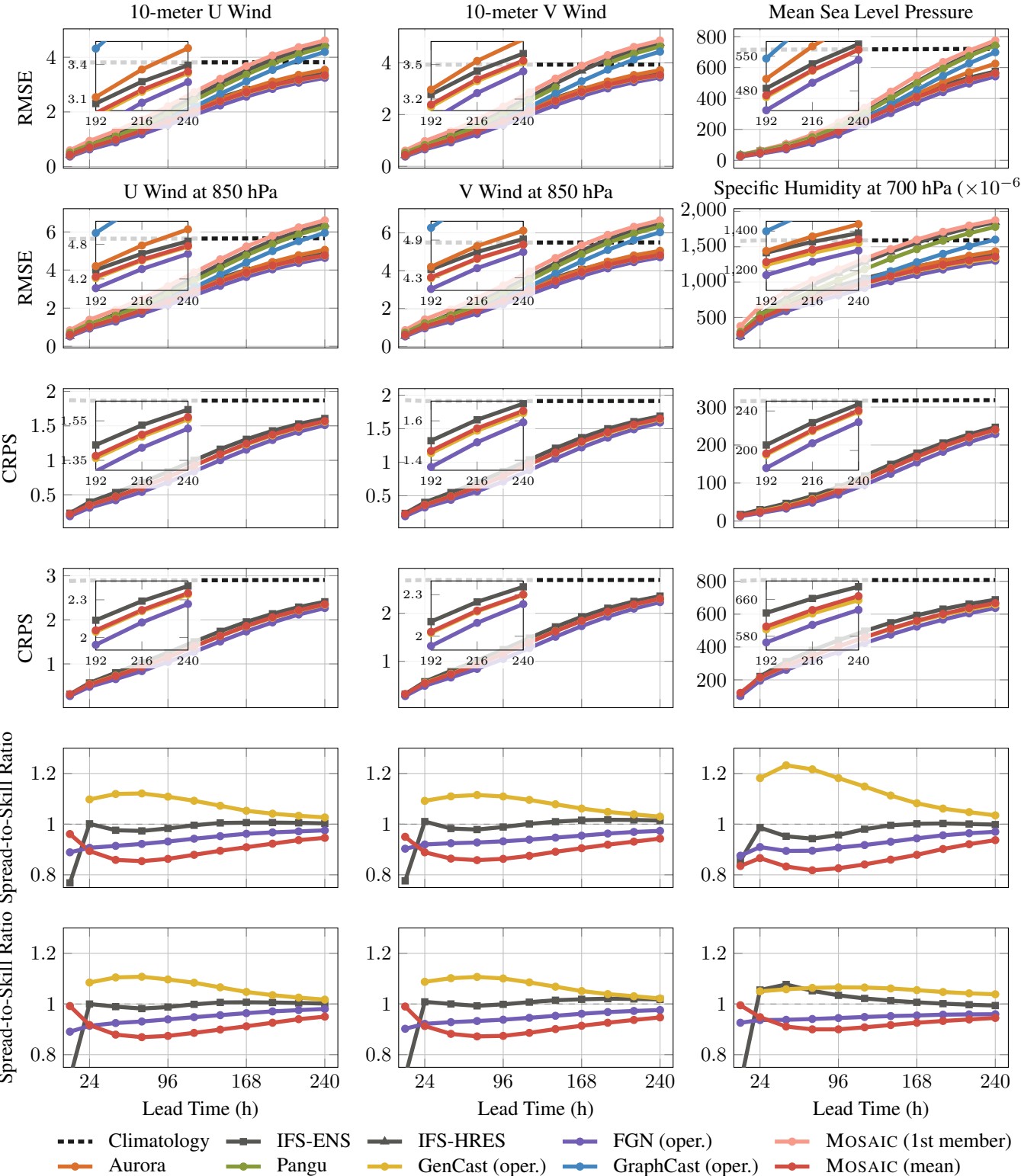

*Figure 7.* RMSE (rows 1–2), CRPS (rows 3–4), and spread-to-skill ratio (rows 5–6) as a function of lead time for additional variables not shown in the main text. Within each metric, the top row shows surface variables (10-meter U-wind, 10-meter V-wind, mean sea level pressure) and the bottom row shows pressure-level variables (U850, V850, Q700). Values close to 1.0 (dashed line) in the SSR rows indicate well-calibrated ensembles.

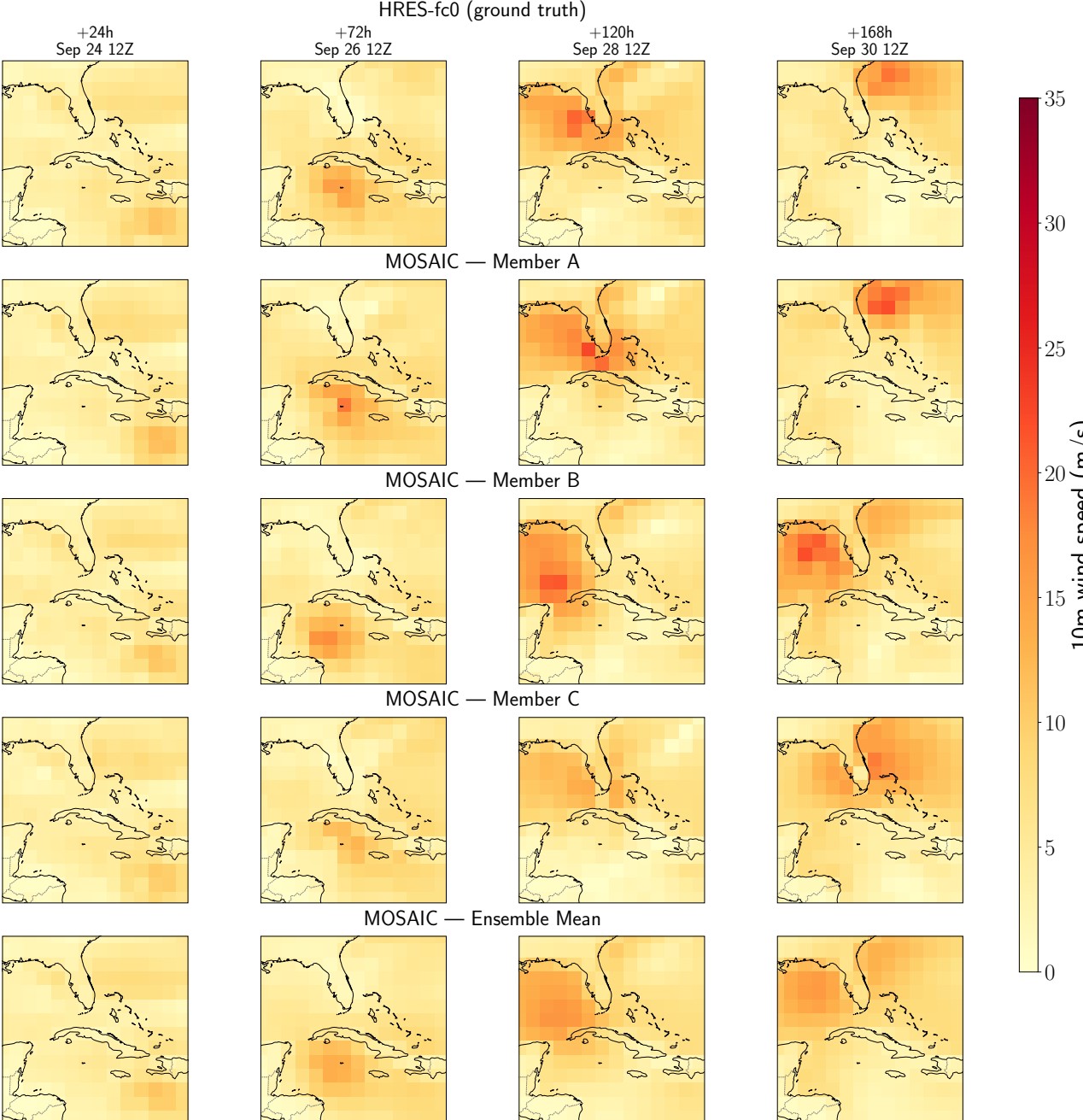

*Figure 8.* Hurricane Ian wind field evolution. Init: Sep 23, 2022 12Z. Columns show lead times (+24 h, +72 h, +120 h, +168 h). Rows: HRES-fc0 ground truth, three individual ensemble members, and ensemble mean (48 members). Variable: 10-meter wind speed (m/s). Region: Gulf of Mexico / Caribbean / southeastern US.

## B.4. Global Mass Conservation

To assess model stability, we evaluate the global mean surface pressure (GMSP) drift over 10-day forecast rollouts. GMSP is computed as the area-weighted (cosine-latitude) global mean of mean sea level pressure. We use the ERA5-trained model with a 24 h prediction step (the same model as the 1.5° benchmark in Section 5) and evaluate over the 2020 test year: 712 initialization dates (00:00 and 12:00 UTC, year 2020) with 48 ensemble members each. Table 6 reports the GMSP drift relative to the initial condition. The maximum mean drift after 10 days is $-0.086$ hPa ($0.009\%$ relative to $\sim 1013$ hPa), confirming that MOSAIC neither systematically creates nor destroys atmospheric mass over extended rollouts. Fig. 9 shows the drift trajectory.

| Lead time | Mean drift (hPa) | Relative | Max member |
|---|---|---|---|
| 24 h | $-0.008 \pm 0.045$ | $0.001\%$ | $< 0.2$ hPa |
| 120 h | $-0.042 \pm 0.112$ | $0.004\%$ | $< 0.5$ hPa |
| 240 h | $-0.086 \pm 0.188$ | $0.009\%$ | $< 1.0$ hPa |

*Table 6.* GMSP drift over 10-day rollouts (2020 test year, 712 init dates $\times$ 48 members).

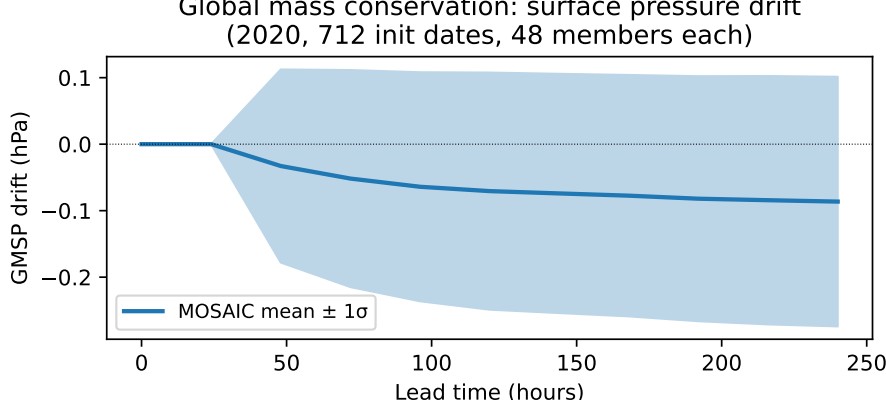

*Figure 9.* GMSP drift (mean $\pm 1\sigma$) over lead time across 34,176 ten-day rollouts. The drift remains below 0.1 hPa throughout, indicating stable mass conservation.

## C. Implementation Details

### C.1. Computational Resources

Table 7 compares MOSAIC with baseline MLWPs. All experiments were conducted on $8\times$ NVIDIA H100 GPUs. Training requires 2 days (16 GPU-days total) using float16 precision throughout.

| Model | Params | Precision | Resolution | Training | Duration |
|---|---|---|---|---|---|
| GraphCast | 36M | fp32 | 0.25° | 32 TPUv4 | 4 weeks |
| GenCast | 57.5M | fp32 | 0.25° | 32 TPUv5 | 5 days |
| FGN | 57.5M | fp32 | 0.25° | 490 TPUv5p/v6e | 3 days* |
| Pangu | 277M | fp32 | 0.25° | 192 V100 | 15 days |
| Aurora | 1,259M | bf16 | 0.25° | 32 A100 | 2.5 weeks |
| MOSAIC | 214M | fp16 | 1.5° | 8 H100 | 2 days |

*Table 7.* Comparison of MOSAIC with baseline MLWPs. MOSAIC achieves competitive performance despite operating at coarser resolution with significantly reduced computational resources.

*Total of 1,470 TPU-days compute (490 TPUs $\times$ 3 days)

### C.2. Data Description

Table 8 lists all input and output variables used by MOSAIC. The model takes as input 4 surface-level variables and 6 pressure-level variables at 13 pressure levels, yielding $4 + 6 \times 13 = 82$ dynamic channels per timestep. Additionally, 3 static fields are concatenated along the feature axis together with Cartesian coordinates $(x, y, z)$ on the unit sphere and 4 sinusoidal time embeddings (sine and cosine of day-of-year and year progress).

| Variable | Type | Unit | Channels |
|---|---|---|---|
| *Surface-level variables (dynamic)* | | | |
| 2-meter temperature | Surface | K | 1 |
| 10-meter U-wind | Surface | m/s | 1 |
| 10-meter V-wind | Surface | m/s | 1 |
| Mean sea level pressure | Surface | Pa | 1 |
| *Pressure-level variables (dynamic)* | | | |
| Geopotential | Pressure | $m^2/s^2$ | 13 |
| Specific humidity | Pressure | kg/kg | 13 |
| Temperature | Pressure | K | 13 |
| U-component of wind | Pressure | m/s | 13 |
| V-component of wind | Pressure | m/s | 13 |
| Vertical velocity | Pressure | Pa/s | 13 |
| *Static variables* | | | |
| Surface geopotential | Static | $m^2/s^2$ | 1 |
| Land-sea mask | Static | 0–1 | 1 |
| Soil type | Static | categorical | 1 |
| **Total dynamic channels per timestep: 82** | | | |

*Table 8.* Complete list of variables used by MOSAIC. Pressure-level variables are provided at 13 levels: {50, 100, 150, 200, 250, 300, 400, 500, 600, 700, 850, 925, 1000} hPa.

**Data normalization.** All dynamic variables are standardized per channel (i.e., per variable and pressure level) using the mean and standard deviation computed over all spatial positions and all training timesteps via Welford's online algorithm. Each input state is normalized as $\hat{x} = (x - \mu)/\sigma$, where $\mu$ and $\sigma$ are the per-channel training statistics. The same normalization is applied to target states: $\hat{y} = (y - \mu)/\sigma$. The model predicts normalized next states directly, and predictions are denormalized at inference time via $x = \hat{x} \cdot \sigma + \mu$. Static variables are normalized independently using their own spatial mean and standard deviation and augmented with Cartesian coordinates $(x, y, z)$ on the unit sphere. During pretraining,

normalization statistics are computed on the ERA5 training split (1979–2018). During finetuning, they are recomputed on the HRES-fc0 analysis data (2016–2021).

**Data splits.** Pretraining uses ERA5 reanalysis from 1979–2018. We evaluate under two benchmarks (see Section 5). For the **1.5° benchmark**, we finetune on ERA5 2007–2019 following Couairon et al. (2024) and test on 2020. For the **0.25° benchmark**, we finetune on HRES-fc0 analysis from 2016–2021 and test on 2022. All data is sampled at 6-hourly temporal resolution.

**Data source and resolution.** All data is obtained from the WeatherBench 2 repository (Rasp et al., 2023) as Zarr archives on Google Cloud Storage. ERA5 reanalysis is provided as `1959-2023_01_10-6h-240x121_equiangular_with_poles_conservative.zarr` and HRES-fc0 analysis as `2016-2022-6h-240x121_equiangular_with_poles_conservative.zarr`. Both datasets are conservatively remapped from their native grids to a $240 \times 121$ equiangular latitude-longitude grid (1.5° resolution) at 6-hourly intervals. No additional regridding or interpolation is applied to the data prior to model input.

### C.3. Model Architecture

Table 9 lists the detailed architecture configuration. The U-Net comprises two coarse-graining stages and a bottleneck, with encoder and decoder layers at each stage.

| Stage | nside | Dim | Heads | Enc. Depth | Dec. Depth | MLP Ratio |
|---|---|---|---|---|---|---|
| Stage 1 | 64 | 768 | 12 | 4 | 2 | 4.0 |
| Stage 2 | 32 | 1024 | 16 | 4 | 2 | 4.0 |
| Bottleneck | 16 | 1280 | 20 | 2 | | 4.0 |
| **Global parameters** | | | | | | |
| Total parameters: 214M | | | | | | |
| GQA ratio: 4, QKV compression ratio: 1 | | | | | | |
| RoPE: enabled ($\theta = 10000$), QK norm: disabled | | | | | | |
| RMSNorm elementwise affine: disabled | | | | | | |
| History steps $T$: 2 (1.5°) / 4 (0.25°), Noise dim: 32, $k$-neighbors: 24 | | | | | | |
| **Block-sparse attention parameters** | | | | | | |
| Block attention size: 1024 | | | | | | |
| Sparse block size: 128 | | | | | | |
| Sparse block count: 24 (stage 1), 12 (stage 2), 4 (bottleneck) | | | | | | |

*Table 9.* Model architecture details.

**Input embedding.** Each spatial location on the latitude-longitude grid constitutes one token. For a history of $T$ consecutive timesteps ($T$=2 for the 1.5° benchmark, $T$=4 for the 0.25° benchmark), the $C_{\mathrm{dyn}}$=82 dynamic channels at each grid point are concatenated along the channel dimension. This vector is further concatenated with the static variables augmented with Cartesian unit-sphere coordinates ($C_{\mathrm{static}}$=6) and a single target-time embedding ($C_{\mathrm{time}}$=4: sine and cosine of day-of-year and year progress for the prediction target time (Bodnar et al., 2025)), yielding a total input dimension of $T \times C_{\mathrm{dyn}} + C_{\mathrm{static}} + C_{\mathrm{time}}$ per token (174 for 1.5°, 338 for 0.25°). A preprocess MLP (Linear $\rightarrow$ RMSNorm $\rightarrow$ SiLU $\rightarrow$ Linear $\rightarrow$ RMSNorm) projects this to the hidden dimension of Stage 1, after which features are interpolated to the HEALPix mesh.

**HEALPix mesh.** The HEALPix grid is generated with the `healpy` library (Zonca et al., 2019) using *nested* pixel ordering, which groups spatially nearby pixels into contiguous memory locations. The resolution parameter $N_{\mathrm{side}}$ determines the total number of pixels as $12N_{\mathrm{side}}^2$: Stage 1 operates at $N_{\mathrm{side}} = 64$ (49,152 pixels), Stage 2 at $N_{\mathrm{side}} = 32$ (12,288 pixels), and the bottleneck at $N_{\mathrm{side}} = 16$ (3,072 pixels). Fig. 10 illustrates the recursive subdivision and the NESTED child indexing.

**Downsampling and upsampling.** Transitions between resolution levels follow the HEALPix quad-tree hierarchy with a downsampling factor of $f$=4. During downsampling (Section 4), $W_x^{\downarrow} \in \mathbb{R}^{d_{\mathrm{out}} \times 4d_{\mathrm{in}}}$ projects the stacked child features

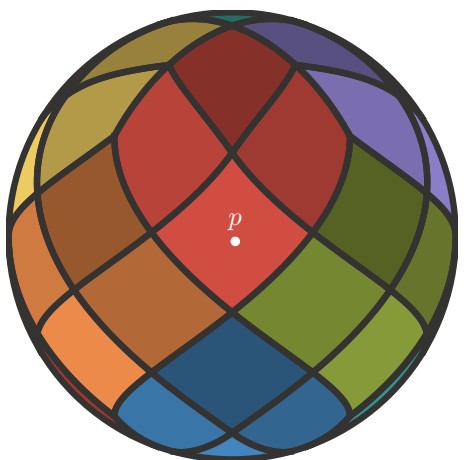 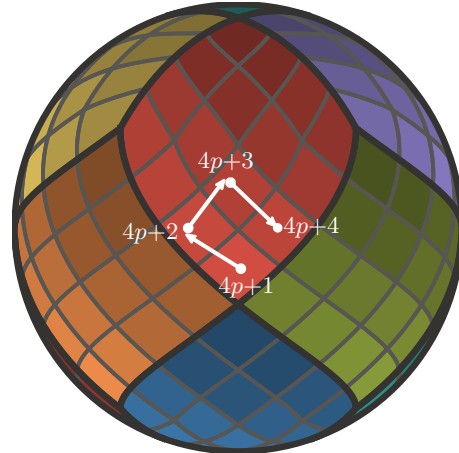

*Figure 10.* HEALPix mesh refinement. Each pixel (left) is subdivided into four children (right), whose indices follow a Z-order curve that keeps spatially close pixels contiguous in memory.

and $W_p^{\downarrow} \in \mathbb{R}^{d_{\text{out}} \times 12}$ provides a position-aware bias from the $4 \times 3$ relative Cartesian coordinates, followed by RMSNorm. During upsampling, $W_x^{\uparrow} \in \mathbb{R}^{4d_{\text{out}} \times d_{\text{in}}}$ expands each coarse pixel to four fine pixels and $W_p^{\uparrow} \in \mathbb{R}^{4d_{\text{out}} \times 12}$ again provides a position-aware bias. The encoder skip connection is added element-wise after the upsampling projection, followed by RMSNorm.

**Output head.** After the final decoder stage, features are normalized via RMSNorm, interpolated from HEALPix back to the latitude-longitude grid (Eq. 4 with source and target reversed), and passed through a postprocess MLP (RMSNorm → Linear → SiLU → Linear). The final linear layer maps to $C_{\text{dyn}} = 82$ output channels with no activation function. The model predicts a normalized next state directly; denormalization recovers physical units at inference time.

**Cross-attention interpolation.** In the interpolation module (Eq. 4), queries are derived from geometry with $W_q \in \mathbb{R}^{d \times 3}$ acting on $L_2$-normalized relative Cartesian positions, while keys and values are derived from RMSNorm-normalized source features via $W_k, W_v \in \mathbb{R}^{d \times d}$. An output projection $W_o \in \mathbb{R}^{d \times d}$ is applied after the attention-weighted sum. All projections are fully learned; neighbor indices and relative positions are fixed after initialization.

**Noise injection details.** The cSwiGLU layer injects noise as an additive bias inside the SwiGLU gate:

$$\text{cSwiGLU}(x, z) = \left( \sigma \left( x W_g + z \right) \odot \left( x W_v \right) \right) W_{out}, \tag{9}$$

where $\odot$ denotes element-wise multiplication and $\sigma(x) = x / (1 + e^{-x})$ is the Swish activation. This induces an effective output projection $W_{out}^{\text{eff}} = \text{diag}(s) \cdot W_{out}$, where the scaling factors $s_i$ are drawn from input-dependent distributions determined by the gate activations $x W_g$. The learned weights $W_{out}$ define the mean structure, while noise $z$ flowing through the input-dependent nonlinearity defines a learned, adaptive covariance: the model learns which features vary across ensemble members based on the input rather than applying uniform noise everywhere.

The noise vector $\mathbf{z} \in \mathbb{R}^{32}$ is sampled once per forward pass from $\mathcal{N}(\mathbf{0}, \mathbf{I})$ and transformed by a learned layer $W_z \in \mathbb{R}^{32 \times 32}$. In each cSwiGLU block (Section 4), a per-layer projection $W_n \in \mathbb{R}^{d_{\text{ff}} \times 32}$ maps $\mathbf{z}$ to the feed-forward hidden dimension. Both $W_z$ and all $W_n$ are initialized with near-zero weights $\sim \mathcal{N}(0, 0.01)$, so that noise injection has negligible effect at the start of training and its influence is learned gradually. The same $\mathbf{z}$ is broadcast to every transformer block across the encoder, bottleneck, and decoder, acting as a global latent variable. Different ensemble members receive independently sampled noise vectors.

## C.4. Training Schedule

Table 10 shows the complete training schedule. Both pretraining and finetuning use a step size of 24 h for the 1.5° benchmark and 6 h for the 0.25° benchmark. Finetuning uses progressively longer autoregressive rollouts in both cases.

| Stage | Steps | AR Length | Learning Rate | Weight Decay | Schedule |
|---|---|---|---|---|---|
| Pretrain | 250k | 1 | 1e-3 | 1e-2 | Cosine (1e-3 → 1e-6), no warmup |
| Finetune 1 | 30k | 1 | 1e-4 | 1e-2 | Cosine decay by 1e-2 |
| Finetune 2 | 10k | 2 | 1e-5 | 1e-2 | Cosine decay by 1e-2 |
| Finetune 4 | 5k | 4 | 5e-6 | 1e-2 | Cosine decay by 1e-2 |
| Finetune 8 | 2.5k | 8 | 1e-6 | 1e-2 | Cosine decay by 1e-2 |
| Finetune 12 | 2.5k | 12 | 1e-6 | 1e-2 | Cosine decay by 1e-2 |

*Table 10.* Training schedule details.

**Optimizer.** We use Muon (Jordan et al., 2024) with momentum $\beta = 0.95$, Nesterov acceleration enabled, 5 Newton-Schulz orthogonalization steps, and no weight decay beyond what is specified per stage in Table 10. The base learning rate is 0.02; per-stage values are listed in Table 10 and follow the cosine schedule described therein.

**Learning rate warmup.** Pretraining uses no warmup. All finetuning stages employ a 500-step linear warmup from $10^{-6} \times \eta$ to the stage-specific learning rate $\eta$, followed by cosine annealing.

**Early stopping.** We apply early stopping based on validation loss at each stage. For pretraining, the criterion is the single-step prediction loss (24 h for the 1.5° benchmark, 6 h for the 0.25° benchmark). For finetuning stage $k$ (with $k$ autoregressive rollout steps), the criterion is the loss on the $k$-th predicted step. The best checkpoint from each stage initializes the next.

**Gradient clipping.** We clip gradient norms to a maximum of 1.0 at every training step across all stages.

**Distributed training.** All experiments use distributed data parallelism (DDP) across 8 NVIDIA H100 GPUs with a per-GPU batch size of 2 (effective batch size 16). Training is performed entirely in float16 precision.

**Weight initialization.** All linear layers are initialized with $\mathcal{N}(0, \sigma)$ where $\sigma = \frac{1}{\sqrt{d_{in}}} \min\left(1, \sqrt{d_{out}/d_{in}}\right)$, and all biases are set to zero. Residual-path layers – the gate portion of SwiGLU ($W_{13b}$), the attention output projection ($W_o$), noise bias projections ($W_n$), noise generator ($W_z$), and upsampling projections – are initialized with $\mathcal{N}(0, 0.01)$ so that residual contributions are near-identity at the start of training.

**Autoregressive rollout during finetuning.** Finetuning stages with $k > 1$ autoregressive steps use the pushforward trick (Brandstetter et al., 2022): the first $k-1$ rollout steps are computed without gradients (i.e., with stopped gradients), and only the final $k$-th prediction step receives gradients. This avoids backpropagation through the full rollout chain, keeping memory requirements constant regardless of the number of rollout steps.

**Training ensemble generation.** During training, we use an ensemble of size $N=2$. At the first autoregressive step, the input state is replicated $N$ times within a single forward pass, and each replica receives an independently sampled noise vector **z**, producing two ensemble members. For subsequent rollout steps, no further branching occurs: each of the two members evolves independently, receiving its own independently sampled noise vector at each step. The two-member ensemble structure is thus maintained throughout the full rollout. The CRPS loss (Section C.5) is computed over the resulting 2-member ensemble at the final rollout step (the only step receiving gradients; see autoregressive rollout paragraph above). At inference, the ensemble size is increased to 48.

## C.5. Loss Function

We use an unbiased (fair) CRPS estimator (Zamo & Naveau, 2018), which for a single variable $y \in \mathbb{R}$ and an ensemble $x^{1:N}$ is given by

$$\mathrm{CRPS}(x^{1:N}, y) := \frac{1}{N} \sum_n |x^n - y|$$
$$- \frac{1}{2N(N-1)} \sum_{n,n'} |x^n - x^{n'}|. \tag{10}$$

The training objective is the latitude-weighted, variable-weighted fair CRPS:

$$\mathcal{L} = \frac{1}{|D|} \sum_{d \in D} \frac{1}{HW} \sum_{h,w} \sum_{i=1}^{C} \alpha_i \, \omega_h \, \mathrm{CRPS}(\hat{x}_{i,h,w,d}^{1:N}, \hat{y}_{i,h,w,d}), \tag{11}$$

where $d$ indexes the batch, $(h, w)$ indexes spatial grid points on the $H \times W$ latitude-longitude grid, $i$ indexes the $C{=}82$ output channels, $\alpha_i$ is the per-channel variable weight, and $\omega_h$ is the latitude weight. Both predictions $\hat{x}$ and targets $\hat{y}$ are in standardized (zero-mean, unit-variance) space.

**Variable-level loss weights.** Following Lam et al. (2023), pressure-level variables are weighted proportionally to their pressure level $p$ (in hPa), normalized by the mean pressure across all 13 levels ($\bar{p} = 463.46\,\mathrm{hPa}$): $\alpha(p) = p/\bar{p}$. This assigns higher weight to lower-tropospheric levels, reflecting their greater meteorological importance for surface weather. Surface variables receive fixed weights. Table 11 lists all weights.

| Variable / Level | $\alpha_i$ |
|---|---|
| *Surface variables* | |
| 2-meter temperature | 1.0 |
| 10-meter U-wind | 0.1 |
| 10-meter V-wind | 0.1 |
| Mean sea level pressure | 0.1 |
| *Pressure levels (shared across all 6 pressure-level variables)* | |
| 1000 hPa | 2.157 |
| 925 hPa | 1.996 |
| 850 hPa | 1.834 |
| 700 hPa | 1.510 |
| 600 hPa | 1.294 |
| 500 hPa | 1.079 |
| 400 hPa | 0.863 |
| 300 hPa | 0.647 |
| 250 hPa | 0.539 |
| 200 hPa | 0.431 |
| 150 hPa | 0.324 |
| 100 hPa | 0.216 |
| 50 hPa | 0.108 |

*Table 11.* Variable-level loss weights $\alpha_i$. All six pressure-level variables share the same per-level weight.

**Latitude weighting.** To account for the convergence of meridians toward the poles, each grid row at latitude $\phi_h$ is weighted proportionally to the area it represents:

$$\omega_h = \frac{\cos(\phi_h)}{\frac{1}{H} \sum_{h'=1}^{H} \cos(\phi_{h'})}, \tag{12}$$

so that the weights average to unity across latitudes.

**Loss normalization.** The loss is computed in standardized space: both model outputs and targets are normalized per channel using the training-set mean and standard deviation (Section C.2). The mean in the loss is taken over all dimensions (batch, latitude, longitude, and channels), with $\alpha_i$ and $\omega_h$ acting as importance multipliers. Training uses mixed-precision (float16) with gradient scaling via PyTorch's `GradScaler`. No additional loss normalization or scaling is applied.

### C.6. Native Sparse Attention Equations

We expand on the three NSA branches summarized in Section 3.

**Compression** Let $\{B_1, \ldots, B_m\}$ be a partition of tokens into $m$ non-overlapping blocks. For each $B_j$, let $\mathbf{K}_j = \{\mathbf{k}_l \mid l \in B_j\}$ denote the set of keys in the block. A block representation is computed via a learnable function $\varphi$:

$$\bar{\mathbf{k}}_j = \varphi(\mathbf{K}_j). \tag{13}$$

Coarse-grained values are obtained as $\bar{\mathbf{v}}_j = \varphi(\mathbf{V}_j)$ with $\mathbf{V}_j = \{\mathbf{v}_l \mid l \in B_j\}$. Coarse-grained attention between query $i$ and each block is then evaluated, with attention scores retained for the selection branch:

$$a_{ij} = \frac{\exp\left(\mathbf{q}_i^\top \bar{\mathbf{k}}_j / \sqrt{d_k}\right)}{\sum_{l=1}^{m} \exp\left(\mathbf{q}_i^\top \bar{\mathbf{k}}_l / \sqrt{d_k}\right)}, \quad \mathbf{o}_i^{CG} = \sum_{j=1}^{m} a_{ij} \bar{\mathbf{v}}_j. \tag{14}$$

The compression branch both captures global context and guides sparsification in the selection branch.

**Selection** For each query $i$, NSA selects the top-$n$ blocks with the highest coarse-grained attention scores, $\mathcal{S}_i = \text{top-}n\,(a_{i,:})$, and computes fine-grained attention over keys and values *within* the selected blocks:

$$\mathbf{o}_i^{FG} = \sum_{j \in \mathcal{S}_i} \sum_{l \in B_j} \frac{\exp\left(\mathbf{q}_i^\top \mathbf{k}_l / \sqrt{d_k}\right)}{Z_i} \mathbf{v}_l \tag{15}$$

where $Z_i = \sum_{j \in \mathcal{S}_i} \sum_{l \in B_j} \exp\left(\mathbf{q}_i^\top \mathbf{k}_l / \sqrt{d_k}\right)$ is the normalizing constant. By operating at full resolution, the selection branch preserves fine-scale detail while capturing long-range interactions.

**Local Attention** The local branch applies standard attention for each query $i$ over keys and values within a sliding window, yielding $\mathbf{o}_i^L$. This frees the compression and selection branches to focus on long-range interactions.

### C.7. Block-sparse attention implementation

**Kernel Design** We implement block-sparse attention in Triton, using the FLA (Yang & Zhang, 2024) implementation as a foundation and following the memory-efficient approach of FlashAttention (Dao et al., 2022). The forward pass loads query blocks into SRAM and streams selected key-value blocks through, computing attention without materializing the full attention matrix. The backward pass computes gradients for keys and values by iterating over all query blocks and only loading those into memory that are connected to a given key-value pair. The original NSA implementation batches query heads sharing the same key-value head, requiring sufficiently large batch sizes[1] for the dot product operations to be executed efficiently on Tensor Cores. In BSA, operating at the block level relaxes this constraint: batching occurs naturally across queries within a block, which at sizes $\geq$128 satisfies tile requirements and enables arbitrary GQA group size.

**Block-sparse attention vs. native sparse attention.** Figure 11 provides a per-component breakdown of BSA and NSA runtime. BSA's block-structured memory access patterns better utilize GPU tensor cores compared to NSA's irregular sparsity patterns, making BSA particularly suitable for high-resolution atmospheric grids. Table 12 lists exact runtimes including dense attention (compiled `flex_attention`); see Fig. 4 for a visual comparison.

### C.8. Computational Efficiency

**Training efficiency.** As detailed in Table 7, MOSAIC requires only 16 GPU-days total, operating entirely in float16 precision. In contrast, GraphCast trains for 4 weeks on 32 TPUv4s, GenCast for 5 days on 32 TPUv5s, and FGN accumulates

---

[1]At least 16 along each dimension in Triton.

| Seq. len. | BSA | NSA | Dense | BSA/Dense |
|---|---|---|---|---|
| 8k | 2.97 | 7.52 | 2.68 | 0.9× |
| 16k | 6.31 | 17.95 | 11.20 | 1.8× |
| 32k | 13.13 | 36.49 | 44.75 | 3.4× |
| 64k | 27.84 | 86.04 | 176.85 | 6.4× |
| 128k | 57.15 | 234.12 | 752.74 | 13.2× |
| 256k | 114.72 | 700.32 | 3,383.69 | 29.5× |
| 524k | 230.68 | 2,115.22 | 14,258.72 | 61.8× |

*Table 12.* Forward pass runtime (ms) for BSA, NSA, and dense attention across sequence lengths. Measured on NVIDIA RTX A4500 (batch=1, 16 heads, head_dim=32). Dense uses compiled `flex_attention`.

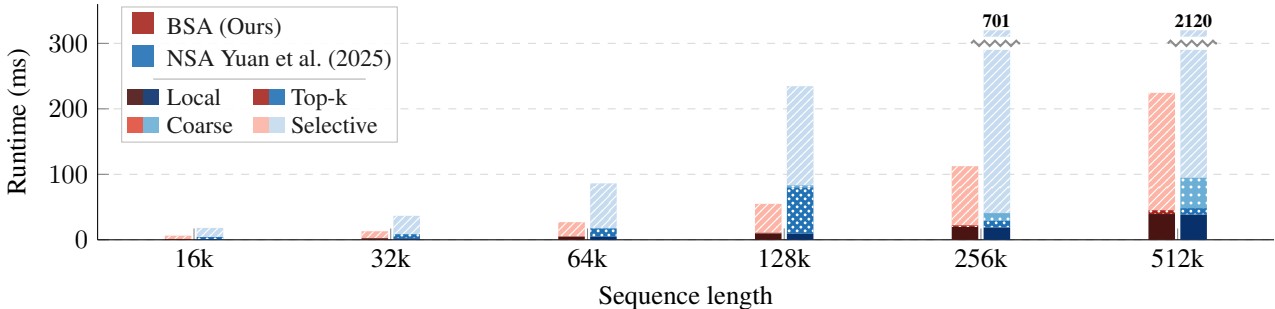

*Figure 11.* Runtime comparison of native sparse attention (NSA) vs block-sparse attention (BSA, ours) on NVIDIA A4500. NSA implementation from Yang & Zhang (2024). BSA achieves consistent speedups across all sequence lengths.

490 TPU-days of compute, all using float32 precision. The combination of lower resolution, the Muon optimizer, float16 precision, and block-sparse attention enables MOSAIC to achieve competitive performance with substantially reduced compute relative to comparable probabilistic models.

**Inference speed.** Table 13 compares inference cost on a single NVIDIA H100 GPU. We report the amortized time per ensemble member per autoregressive step (**s/member/step**), measured from the largest ensemble that fits in memory, together with the peak GPU memory for that configuration. MOSAIC produces a 24-member, 10-step (240 h) ensemble in 11.60 s (0.048 s/member/step), on par with the deterministic Stormer while generating full probabilistic forecasts. Among other ensemble-capable models at comparable resolution, ArchesWeather-Mx4 is similarly fast but limited to 4 members, while ArchesWeatherGen requires 0.986 s/member/step due to its diffusion-based sampling.

| Model | Res. | Params | Prec. | s/mem./step | Peak (GB) |
|---|---|---|---|---|---|
| MOSAIC (Ours) | 1.5° | 214M | fp16 | 0.048 | 3.26 |
| ArchesWeatherGen | 1.5° | 384M | fp16 | 0.986 | 2.40 |
| ArchesWeather-Mx4 | 1.5° | 339M | fp16 | 0.058 | 2.08 |
| Stormer | 1.4° | 469M | fp16 | 0.062 | 11.24 |
| NeuralGCM[†] | 1.4° | 12M | fp32 | 2.182 | 2.18 |
| GraphCast[†] | 0.25° | 36M | bf16 | 42.57 | 36.55 |
| GenCast[†§] | 0.25° | 57M | fp32 | 197.65 | 33.34 |
| Pangu-Weather[‡] | 0.25° | – | fp32 | 0.66 | 66.69 |

*Table 13.* Inference benchmarking on a single NVIDIA H100 GPU. **s/mem./step** is the wall-clock time per ensemble member per autoregressive step, computed from the largest ensemble run that fits in memory (24 members for MOSAIC, ArchesWeatherGen, and NeuralGCM; 4 for ArchesWeather-Mx4; 1 for the rest). `torch.compile` is enabled for PyTorch models where applicable.

[†] JAX models. [§] GenCast uses diffusion-based sampling (1 noise level).
[‡] ONNX Runtime (no `torch.compile`).

## C.9. Evaluation Protocol

**Inference ensemble generation.**    At inference, we generate a 48-member ensemble following the same branching scheme as during training (Section C.4): all members share the same initial condition and diverge at the first autoregressive step through independently sampled noise vectors. Each member then evolves autoregressively with a single noise sample per step.

**Baseline results.**    All baseline scores reported in this paper are obtained from the WeatherBench 2 public evaluation framework (Rasp et al., 2023). Before computing metrics, WeatherBench 2 first-order conservatively regrids all forecasts and ground truths to 1.5° resolution, ensuring a common evaluation grid. Since MOSAIC operates natively at 1.5°, no additional regridding is required for our model. We do not re-run baseline models; we use their publicly available forecast outputs evaluated through WeatherBench 2.

## C.10. Reproducibility

**Software stack.**    All experiments are implemented in PyTorch 2.8 with CUDA 12.8. Custom block-sparse attention kernels are written in Triton 3.6. Key dependencies include `einops` (tensor rearrangement), `healpy` (Zonca et al., 2019) (HEALPix grid generation), `scikit-learn` (BallTree for neighbor lookup), `xarray` and `zarr` (data loading from WeatherBench 2), and `wandb` (experiment tracking). Spectral analysis uses `pyshtools` and `scipy`.

**Code and data availability.**    Training and evaluation data are publicly available through the WeatherBench 2 repository (Rasp et al., 2023) on Google Cloud Storage.

## C.11. Aliasing in Compressive Weather Models

We elaborate on the second failure mode of spectral degradation: high-frequency artifacts arising from compressive encoding. The mechanism is aliasing, which has been studied in neural operators (McCabe et al., 2023; Fanaskov & Oseledets, 2022) and in vision transformers (Michaeli & Soudry, 2025). Here we summarize how it applies to weather models built on HEALPix and what we observe in our compression ablation.

**Nyquist limit on the HEALPix grid.**    The HEALPix mesh at resolution $N_{\text{side}}$ contains $12N_{\text{side}}^2$ pixels. MOSAIC enters the U-Net at $N_{\text{side}}=64$ (49,152 pixels), MOSAIC-C at $N_{\text{side}}=32$ (12,288 pixels). Halving $N_{\text{side}}$ halves the number of representable spatial modes and therefore halves the Nyquist wavenumber of the latent grid. Features at finer scales than the latent Nyquist cannot be represented by the spatial layout of the coarse grid alone.

**Aliasing mechanism.**    When the input is projected onto the coarse grid and then passed through pointwise nonlinearities, energy from frequencies above the coarse Nyquist limit folds back onto resolved frequencies (McCabe et al., 2023; Fanaskov & Oseledets, 2022). Once folded, this energy is indistinguishable from the resolved modes and is carried through the remaining nonlinear operations. Upon decoding back to the native grid, it re-emerges as spurious power at fine scales. The resulting spectral signature is an upward deviation of the model-to-reference ratio near the Nyquist limit – the opposite of the damping produced by deterministic training.

**Role of learnable interpolation.**    The Nyquist argument above describes a hard limit that applies when fine-scale information has nowhere to be stored other than across spatial pixels. MOSAIC and MOSAIC-C both project to the HEALPix mesh via the learnable cross-attention interpolation of Section 4, which augments each coarse pixel with a multi-channel feature vector. Fine-scale variation can therefore be partially absorbed across the feature dimension rather than the spatial dimension, softening the hard Nyquist cut-off. As Fig. 2(a) shows, this offsets the onset of aliasing well beyond the formal $N_{\text{side}}=32$ Nyquist wavenumber, but does not eliminate it: in MOSAIC-C, the ratio still diverges upward in the highest resolved wavenumbers, where the channel capacity is no longer sufficient to encode the missing spatial detail.

**Empirical signature.**    Fig. 2(a) reports kinetic energy spectral ratios at 24 h, aggregated over the 2020 test year (720 initial conditions, 16 ensemble members for probabilistic models). MOSAIC is stable across the spectrum, whereas MOSAIC-C grows above 1.0 in the last wavenumbers before Nyquist. A qualitatively similar bump is visible for GENCAST in the same figure, which also uses a coarse latent grid. Together with the 3.9% nRMSE degradation of the spatial-compression

ablation (Table 2 g), this evidence indicates that compressive encoding induces a distinct architectural failure mode of spectral degradation, alongside the statistical damping discussed in the main text.

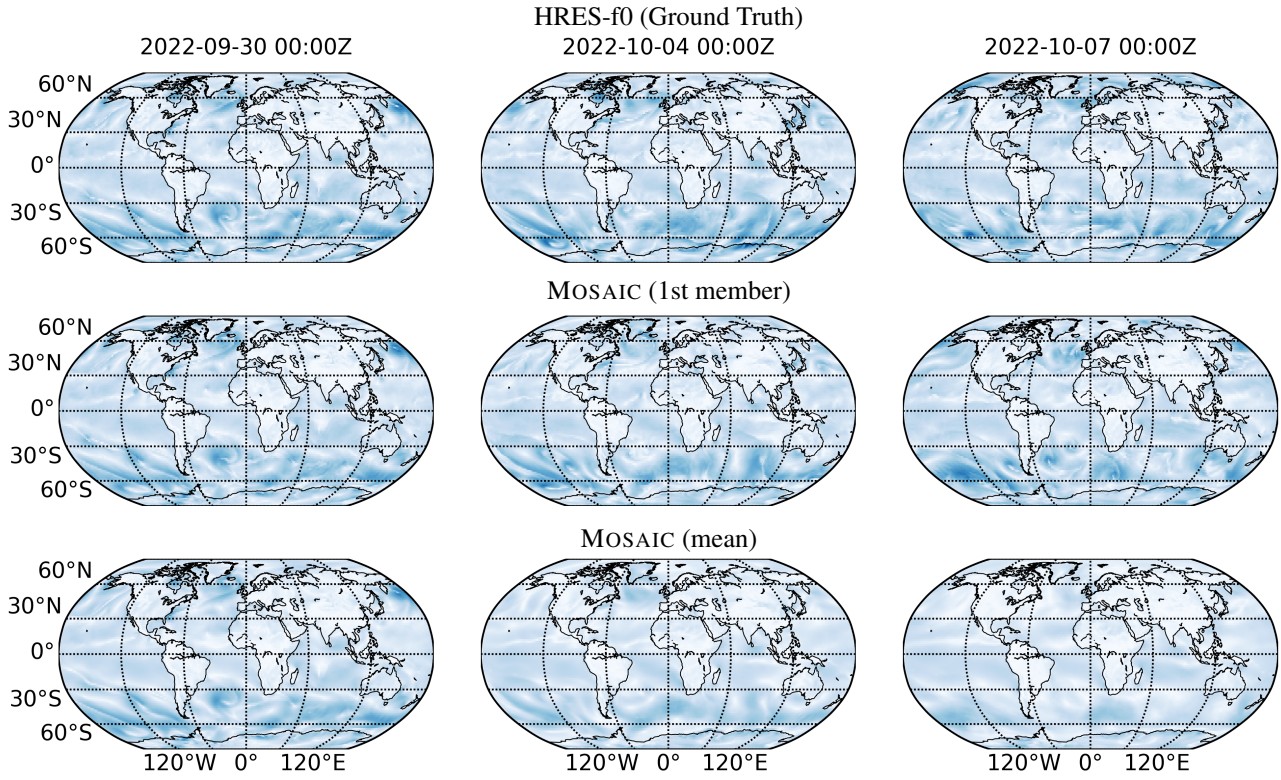

*Figure 12.* Forecast rollout trajectories showing 10-day evolution of wind speed fields at 850 hPa.

## D. Limitations and Future Work

MOSAIC operates at $1.5°$ ($\sim$166 km), which cannot resolve mesoscale phenomena such as tropical cyclone inner-core structure or individual severe thunderstorms. The linear cost and global receptive field of block-sparse attention make scaling to finer grids a natural next step, particularly at $0.25°$ ($\sim$700k tokens), where the architectural advantages over compression-based models should be most pronounced. We estimate that the training would require $\sim$20,000 H100 GPU-hours, which is comparable to existing $0.25°$ MLWPs. Beyond resolution scaling, MOSAIC's efficiency (12 s for a 24-member, 10-day forecast on a single GPU) opens the door to real-time ensemble nowcasting, rapid ensemble data assimilation cycles, and operational decision support. Finally, sparse attention over heterogeneous token sets could enable direct ingestion of sparse observations alongside gridded state tokens, potentially bypassing the costly analysis step that current MLWPs require.

## E. Extended Related Work

### E.1. Compression Effect on Expressivity

The effect of compressive encoding on model expressivity is well studied in computer vision, where patchification, the core tokenization strategy of vision transformers (Dosovitskiy et al., 2020), reduces computational cost by compressing spatial information. Wang et al. (2025) demonstrate irreversible information loss caused by patchification and show that test loss declines consistently as patch size decreases, reaching optimal performance at $1 \times 1$ patches. Moreover, reducing patch size is more beneficial than increasing model parameters, indicating that added capacity cannot compensate for compression-induced information loss. The same pattern holds in weather forecasting, where Nguyen et al. (2023) show that decreasing patch size consistently improves forecast accuracy.

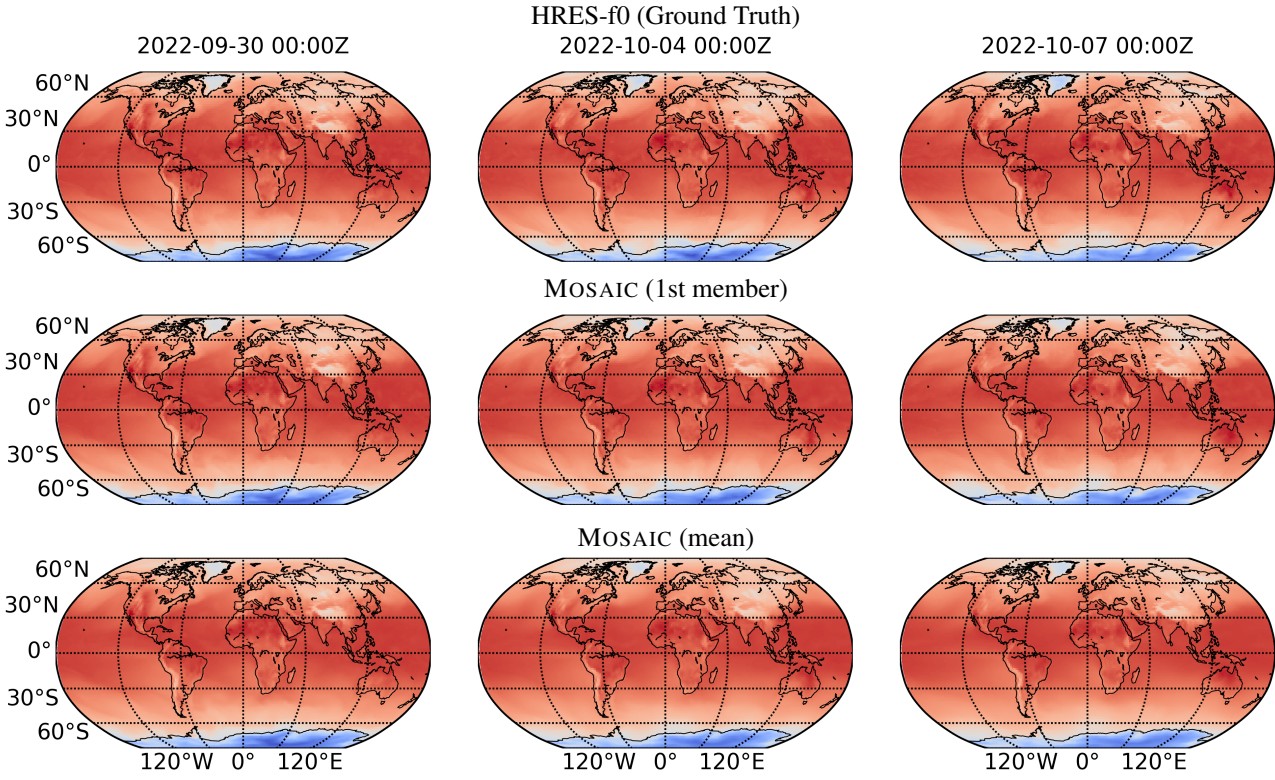

*Figure 13.* Forecast rollout trajectories showing 10-day evolution of surface temperature fields.

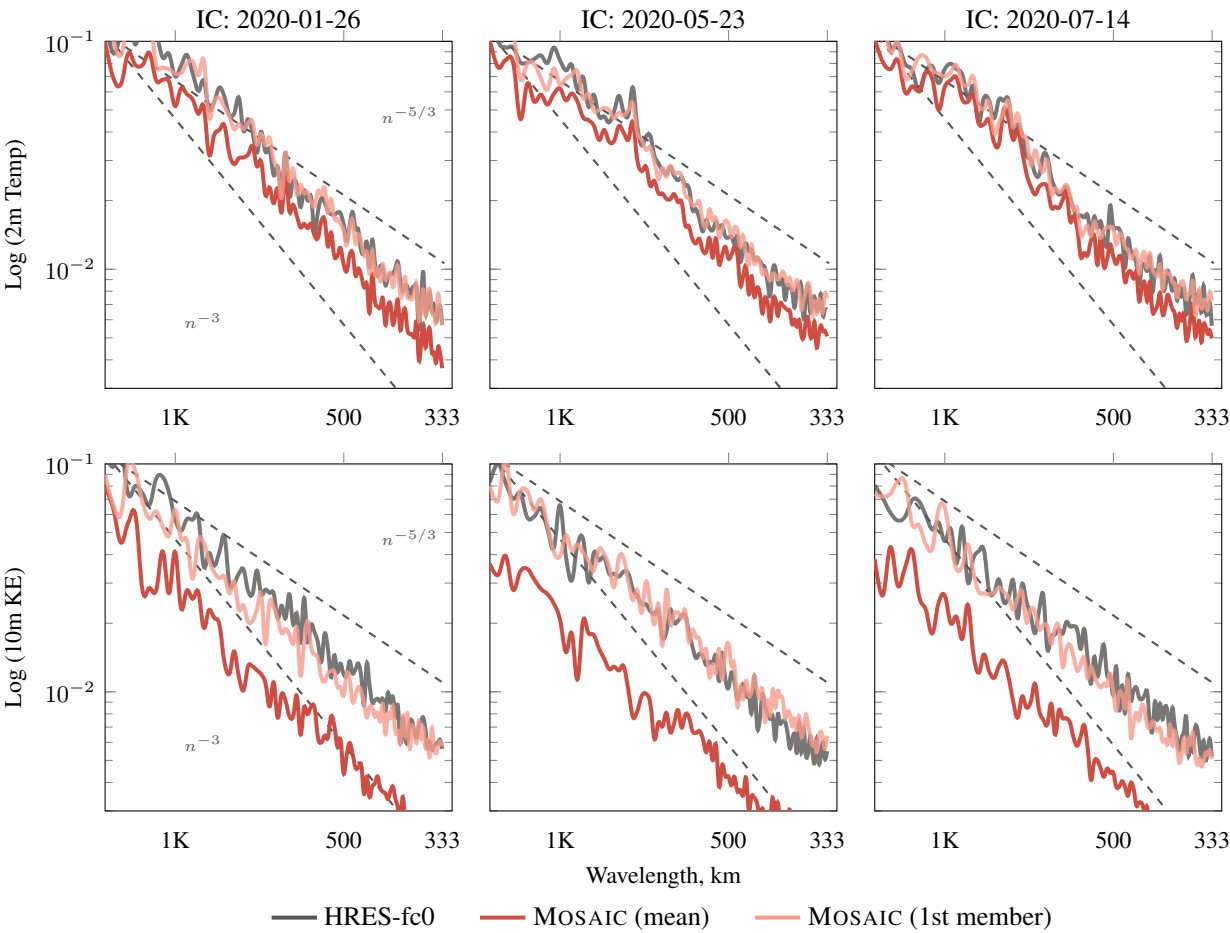

*Figure 14.* Examples of global temperature spectra at 2-meter height (top row) and kinetic energy spectra at 10-meter height (bottom row) for 10-day MOSAIC forecasts compared to HRES-fc0 0.25° ground truth, shown for multiple initial conditions (all start at 00:00 UTC).

For atmospheric data, where fine-scale spatial structure is associated with extreme weather events, this information loss is especially consequential. We avoid the compress-first design: rather than projecting onto a coarse mesh before any spatial mixing, we capture spatial interactions at native resolution via block-sparse attention, with coarsening applied only after the first encoder stage.

### E.2. Aliasing in Neural Architectures

A complementary line of work studies how compressive architectures interact with the frequency content of their inputs. McCabe et al. (2023) analyze aliasing in autoregressive neural operators, showing that nonlinear activations generate modes above the Nyquist limit of the representation, which then fold back onto resolved frequencies and accumulate over rollouts. Fanaskov & Oseledets (2022) formalize this error and propose representations that avoid it. The same phenomenon arises in vision transformers, where Michaeli & Soudry (2025) identify patchification as a source of aliasing and study anti-aliasing strategies. These results frame compressive encoding as a limiting factor in frequency representability, which is especially consequential for weather forecasting, where extreme events manifest at fine spatial scales.

