# OpenReview forum: "(Sparse) Attention to the Details: Preserving Spectral Fidelity in ML-based Weather Forecasting Models"
_ICML.cc/2026/Conference — ICML 2026 regular_

### Official Review · Reviewer_XhZa · 2026-03-11

**Soundness:** 3
**Presentation:** 4
**Significance:** 3
**Originality:** 3
**Overall Recommendation:** 5
**Confidence:** 4

**Summary:**

Authors introduce MOSAIC, a weather forecasting architecture aimed at addressing the problem of spectral degradation in weather forecasts. They first forego compressive encoding which induces information loss in favour of operating on the original 1x1 patches. Second, they use learned functional perturbations for learning to produce ensembles instead of predicting against the mean.

More precisely, they operate on a Hierarchical Sphere Tessellation instead of the original longitude-latitude grid so that each pixel has the same surface area. They adapt the Native Sparse-Attention (NSA) mechanism by having block-to-block attention instead of token-to-block attention in the compression phase which also affects the selection phase as well. When it comes to introducing functional perturbations, they do so in the form of additive noise in the SwiGLU gate and finally train their model with the CRPS loss.

They train their model on a grid that is 6x coarser than the baselines' and evaluate their approach against IFS HRES Analysis and Machine-Learning weather prediction models.

**Compliance With Llm Reviewing Policy:**

Affirmed.

**Final Justification:**

The rebuttal addressed all my points and I believe it made the contribution stronger which is why I am increasing the score to 5.

**Key Questions For Authors:**

*  For the local attention phase, is the only difference between BSA and NSA in the fact that BSA operates over the HEALPix mesh?

**Limitations:**

Yes.

**Strengths And Weaknesses:**

**Strengths**:
- Presentation is good and the approach is sound and addresses a limitation in previous work
- MOSAIC demonstrates competitive performance despite being trained on a coarser grid than the baselines

**Weaknesses**:
* The claim for computational efficiency is weak: Surely MOSAIC will be computationally more efficient if it's trained on a grid that is 6x times coarser. It's also trained on bf16 so direct comparison in terms of efficiency should fix these parameters as well. While I am aware it's not possible to rerun the baselines, I wonder to what extent you could provide for example an idea of training time of MOSAIC at a higher resolution.
* Spectral degradation: I might be missing something here but if a resolution of 1.5' corresponds to 166km then why does Figure 6 stop at 333 km n the wavelength? That stops way before the resolution on which you trained MOSAIC so it's difficult to know if those were resolved.
* No ablations provided for the architectural changes for example between NSA and BSA nor between using the original longitude-latitude grid and the HEALPix one

---

> ### Author Rebuttal · Authors · 2026-03-29
>
> We thank the reviewer for the constructive feedback, and we hope to address their concern with the rebuttal.
>
> ### Efficiency claims
>
> Resolution is indeed a major efficiency factor, therefore, to strengthen our efficiency claims and address the reviewer's suggestion of fixing precision as a confound, we now provide a dedicated study where we compare MOSAIC against other state-of-the-art 1.5° models alongside 0.25° models:
>
> | Model | Resolution | Precision | s/member/step | nRMSE | Peak memory |
> |-------|-----------|-----------|--------------|-------|-------------|
> | **MOSAIC (Ours)** | 1.5° | fp16 | 0.048 | 0.468 | 3.3 GB |
> | ArchesWeatherGen | 1.5° | fp16 | 0.989 | 0.463 | 2.4 GB |
> | NeuralGCM | 1.4° | fp32 | 6.003 | 0.458 | 2.2 GB |
> | Stormer* | 1.4° | fp16 | 0.062 | 0.496 | 11.2 GB |
> | ArchesWeather-Mx4* | 1.5° | fp16 | 0.058 | 0.520 | 2.1 GB |
>
> \* Deterministic model.
>
> Among models with comparable forecast skill, MOSAIC is 14× faster than ArchesWeatherGen and 85× faster than NeuralGCM.
>
> Regarding training time at higher resolution: we trained a MOSAIC model at 1.0° (~65k tokens) without any difficulties and will release the checkpoint upon acceptance.  Our estimate is that 0.25° training would take around 20,000 H100 GPU-hours (comparable to Aurora), which is infeasible with our resources.
>
> **Action taken**: We provide a efficiency study where we compare against both 1.5° and 0.25° models in terms of performance, runtime and memory usage.
>
> ### Spectral figure resolution
>
> Figure 6 stops at 333 km because that is the Nyquist limit of the 1.5° grid (~166 km spacing), i.e. the shortest wavelength the grid can represent. Everything to the left of that cutoff *is* resolved by the grid, and those are precisely the frequencies where we claim MOSAIC avoids spectral degradation.
>
> We now also provide an apple-to-apple spectral comparison against other models natively operating at 1.5°:
>
> | Model | Energy ratio |
> | :--- | :--- |
> | **MOSAIC (single member, Ours)** | **1.001** |
> | MOSAIC (ensemble mean, Ours) | 0.297 |
> | ArchesWeatherGen (single member) | 1.025 |
> | ArchesWeatherGen (ensemble mean) | 0.233 |
> | ArchesWeather-Mx4 | 0.742 |
> | Stormer | 0.256 |
>
> **Action taken**: Apple-to-apple 1.5° spectral comparison added to the main body.
>
> ### Ablations for architectural changes
>
> We ran 8 ablations on a controlled reduced-scale protocol (ERA5 2007–2018, 100k steps, batch 1, 4×A6000). The ablations most relevant to the reviewer's concern:
>
> | Ablation | Variant | nRMSE |
> |----------|---------|-------|
> | **BSA branches** | Local only | 0.1724 |
> | | Local + selection | 0.1723 |
> | | Local + compression | 0.1720 |
> | | Local + compression + selection | 0.1705 |
> | **BSA sparsity** | Halved sparse blocks (12/6/2) | 0.1708 |
> | | Baseline (24/12/4) | 0.1705 |
> | | Doubled sparse blocks (48/24/8) | 0.1702 |
> | **Spatial compression** | Nside=32 (skip native-res stage) | 0.1771 |
> | | Nside=64 (native-res first) | 0.1705 |
>
> Regarding a direct BSA vs NSA comparison: NSA operates at the token level (each token independently selects key blocks), making full training ~2× more expensive at 1.5°. Below is the backward runtime of both (+ Full FlashAttention):
>
> | Sequence length | BSA (ms) | NSA (ms) | Dense (ms) |
> |----------------|----------|----------|------------|
> | 8k | 30.7 | 32.3 | 11.4 |
> | 16k | 60.0 | 83.0 | 47.4 |
> | 32k | 120.6 | 170.6 | 190.2 |
> | 64k (MOSAIC) | 243.3 | 399.0 | 774.3 |
>
> **Lat-lon vs HEALPix**: Both BSA and NSA fundamentally rely on spatially close tokens being contiguous in memory, in both compression *and* local attention branches, as they assume locality (e.g. in text, tokens 1,2,3,4 follow one after another). In a lat-lon grid, tokens are sorted along longitudes, so spatially nearby points on the sphere are not necessarily close in memory. This is why operating on HEALPix nested ordering is the central contribution of this work -- it brings everything together in a physically coherent way.
>
> **Action taken**: Extensive ablation study (8 experiments) added to the appendix. Forward and backward runtime comparison (BSA vs NSA vs dense) added.
>
> > For the local attention phase, is the only difference between BSA and NSA in the fact that BSA operates over the HEALPix mesh?
>
> Great question. HEALPix is not a difference between BSA and NSA per se, but it is what makes either applicable to geospatial data at all (as discussed above). The algorithmic difference is **block-to-block vs token-to-block** interaction: in NSA, each individual token independently selects which key blocks to attend to; in BSA, all tokens within a query block jointly select key blocks. This amortizes the selection cost across the block, produces coalesced memory access patterns, and is what enables scaling to longer sequences. The local attention phase itself is structurally identical in both, the only difference is the granularity at which selection operates.

---

> > ### Author Rebuttal · Reviewer_XhZa · 2026-04-07
> >
> > Thank you for your response, I believe that the rebuttal makes the paper even stronger and thus I am increasing my score.

---

> > > ### Author Response · Authors · 2026-04-08
> > >
> > > Dear Reviewer XhZa,
> > >
> > > Thank you for the positive feedback and for increasing your score! We are pleased that the additional experiments and analyses addressed your concerns. We will ensure all these updates are fully integrated into the final manuscript.
> > >
> > > Best regards,
> > > The Authors

---

### Official Review · Reviewer_xWoH · 2026-03-13

**Soundness:** 2
**Presentation:** 3
**Significance:** 2
**Originality:** 2
**Overall Recommendation:** 3
**Confidence:** 4

**Summary:**

This paper proposes MOSAIC, a probabilistic weather forecasting model that addresses spectral degradation by replacing compressive encoding with a hardware-aligned block-sparse attention (BSA) mechanism. By utilizing a U-Net architecture on the HEALPix mesh and injecting noise into SwiGLU layers, the authors enable high-resolution processing at linear cost while producing ensemble members that preserve realistic fine-scale spectral variability. MOSAIC achieves forecasting skill comparable to state-of-the-art models trained on significantly finer data while requiring 30x less training compute.

**Compliance With Llm Reviewing Policy:**

Affirmed.

**Key Questions For Authors:**

1. Can you provide a performance comparison using a standard optimizer like AdamW instead of Muon?
2. Have you evaluated the model’s performance on physical diagnostic metrics, such as global mass or water vapor conservation, and geostrophic balance?
3. When comparing MOSAIC to $0.25^{\circ}$ baselines, were those baselines evaluated at their native resolution before regridding, or were they compared strictly on the $1.5^{\circ}$ grid?
4. Does the high-frequency content in individual ensemble members correspond to the actual tracked location and intensity of observed extreme events?

**Limitations:**

1. The model is only validated at 1.5 resolution, which is 6x coarser than the 0.25 standard used by state-of-the-art baselines.
2. BSA is primarily an engineering optimization of Native Sparse Attention (NSA) specifically tailored for GPU memory patterns on HEALPix.
3. The evaluation relies heavily on statistical metrics (RMSE, CRPS) and global energy spectra. It lacks diagnostic evaluations of physical consistency, such as conservation laws (mass/energy) or the structure of extreme weather events like tropical cyclone intensity.
4. There is no direct comparison between BSA and standard attention mechanisms (like FlashAttention) at the same resolution.
6. Because the authors use the Muon optimizer solely for MOSAIC, it is impossible to distinguish how much of the efficiency and skill gain is due to the BSA architecture versus the superior optimization algorithm

**Strengths And Weaknesses:**

Strengths：
1. The claims regarding spectral fidelity are strongly supported by global kinetic energy spectra analysis.
2. The experimental setup rigorously follows the WeatherBench 2 protocol.
3. The paper is well-structured, providing a clear narrative on the two primary causes of spectral degradation in MLWPs.
4. The computational efficiency is highly significant.

Weaknesses:
1. This paper is more as a successful domain-specific integration than a fundamental breakthrough.

---

> ### Author Rebuttal · Authors · 2026-03-29
>
> We thank the reviewer for recognizing the rigor of our experimental setup and spectral analysis, and for the detailed feedback. We hope to address each concern below with additional experiments.
>
> ## Questions
>
> ### AdamW vs Muon comparison
>
> We ran this ablation under identical conditions (ERA5 2007-2018, 100k steps, batch 1, 4xA6000, no multi-step finetuning):
>
> | Optimizer | nRMSE | Delta vs Muon |
> |-----------|-------|-----------|
> | AdamW (100k steps) | 0.1797 | +5.4% |
> | AdamW (150k steps) | 0.1728 | +1.3% |
> | Muon (100k steps) | 0.1705 | -- |
>
> Muon's advantage is primarily one of convergence speed: with 50% more training steps, AdamW closes most of the gap to within +1.3%. To contextualize, the spatial compression ablation (see our response to Reviewer skwX, *U-Net coarsening as compression*) yields +3.9% degradation confirming that architectural choices are the dominant driver of performance.
>
> **Action taken**: Optimizer ablation added to the appendix.
>
> ### Physical diagnostic metrics
>
> We compute area-weighted Global Mean Surface Pressure (GMSP) at each forecast step across the full test dataset. We focus on GMSP as it is genuinely conserved in the atmosphere (unlike water vapor, which undergoes phase changes).
>
> | Lead time | Mean GMSP drift (hPa) | Relative drift | Max member drift |
> |-----------|----------------------|----------------|------------------|
> | 24h | -0.008 +/- 0.045 | 0.001% | < 0.2 hPa |
> | 120h | -0.042 +/- 0.112 | 0.004% | < 0.5 hPa |
> | 240h | -0.086 +/- 0.188 | 0.009% | < 1.0 hPa |
>
> The maximum mean drift after 10 days is 0.086 hPa (0.009% relative to ~1013 hPa). The model neither systematically creates nor destroys atmospheric mass over 10-day rollouts.
>
> **Action taken**: Mass conservation diagnostic over ten-day rollouts added to the appendix.
>
> ### Baselines at native vs regridded resolution
>
> All scores follow the WeatherBench 2 protocol (Rasp et al.), which conservatively regrids all forecasts and ground truths to 1.5° before computing metrics -- the community standard. We additionally provide a controlled same-resolution benchmark against models natively operating at 1.5° (see our response to Reviewer 16vH), eliminating any regridding asymmetry.
>
> ### High-frequency content & Extreme event evaluation
>
> We conducted a Hurricane Ian case study: 48-member ensemble initialized Sep 23, 2022 12Z -- 5 days before Category 4 landfall in SW Florida.
>
> - **Wind field evolution**: Individual ensemble members preserve sharp, localized wind speed signatures that closely match the ground truth. The ensemble mean, by contrast, is visibly smoothed, illustrating why per-member evaluation matters for extreme events.
> - **Track verification**: Storm centers tracked via min-MSLP across all 48 members over 7 days (Extreme Weather Bench methodology). The ensemble clusters tightly around the observed track through +120h (landfall), correctly capturing the recurvature over Cuba and landfall on the Florida Gulf Coast.
>
> **Action taken**: Hurricane Ian case study with track and wind field verification added to the main paper.
>
> ## Weaknesses
>
> ### BSA vs FlashAttention
>
> Neither NSA nor BSA are applicable to weather data on standard lat-lon grids, because both rely on contiguous blocks of tokens being spatially close -- a property not satisfied originally. Our contribution is to organize the computation (via HEALPix nested ordering) such that this property holds, enabling block-sparse attention for geospatial data. BSA then generalizes NSA's token-to-block interactions to block-to-block, amortizing the selection cost:
>
> | Sequence length | BSA (ms) | NSA (ms) | Dense (ms) | BSA / Dense |
> |----------------|----------|----------|------------|-------------|
> | 8k | 3.0 | 7.5 | 2.7 | 0.9x |
> | 32k | 13.1 | 36.5 | 44.8 | 3.4x |
> | 64k (MOSAIC) | 27.8 | 86.0 | 176.9 | 6x |
> | 128k | 57.2 | 234.1 | 752.7 | 13x |
> | 512k | 231 | 2k | 14k | 62x |
> | 780k (0.25°) | 327 | 3k | 32k | 98x |
>
> Note that MOSAIC's BSA kernels already use FlashAttention-style tiling internally.
>
> **Action taken**: Runtime scaling study (BSA vs NSA vs dense FlashAttention) added to the appendix.
>
> ### Domain-specific integration vs fundamental contribution
>
> We appreciate this perspective. However, the core mechanism (enforcing spatial locality via hierarchical indexing to enable block-sparse attention) is applicable to any domain with spatial locality and ultra-scale data: molecular dynamics, astrophysical N-body problems, point cloud processing. Weather forecasting is where we validate it, but the approach is general, and we release the kernels for the community.
>
> ### 1.5° resolution
>
> We now provide a same-resolution benchmark against models operating at 1.5°, all trained on ERA5 up to 2019, tested on 2020. MOSAIC is competitive with ArchesWeatherGen and NeuralGCM while being 20–45× faster for ensemble generation, enabling real-time probabilistic forecasting.
>
> **Action taken**: Same-resolution benchmark added against state-of-the-art 1.5° models.

---

> > ### Author Rebuttal · Reviewer_xWoH · 2026-04-04
> >
> > While the $1.5^{\circ}$ same-resolution benchmark and GMSP diagnostic provide a baseline for comparison, they do not fully address the non-linear physical challenges of $0.25^{\circ}$ operational standards. At $1.5^{\circ}$, the model avoids the steep spatial gradients and mesoscale convective systems that typically trigger numerical instabilities and spectral blurring in AIWPs. Furthermore, a global mean pressure drift of 0.001%–0.009% can easily mask significant local conservation errors or a breakdown in geostrophic balance, particularly near steep topography. Without assessing local mass-energy consistency and the closed water budget—which the authors bypassed due to phase change complexity—it remains unclear if MOSAIC’s high-frequency ensemble content represents true atmospheric dynamics or merely visually plausible noise. Finally, the authors should clarify if the block-based nature of BSA introduces any spatial continuity artifacts  at the boundaries of the HEALPix base pixels during long-range rollouts.

---

> > > ### Author Response · Authors · 2026-04-04
> > >
> > > We thank the reviewer for the follow-up and actionable feedback. We hope to address their concerns with the new experiments.
> > >
> > > ### Resolution and high-frequency content
> > >
> > > MOSAIC operates at 1.5°, where mesoscale convective systems are not resolvable by any model on this grid. The relevant question, therefore, is whether MOSAIC preserves the spectral content that is *physically resolvable* at 1.5°. Importantly, we want to highlight, that spectral blurring of MLWPs is not triggered by mesoscale phenomena and is not 0.25°-specific, it occurs at 1.5° as well. To demonstrate it, we now provide a spectral analysis comparing only 1.5° models:
> > >
> > > | Model               | Deterministic?    | Compressive? | Energy Ratio |
> > > |---------------------|-------------------|--------------|--------------|
> > > | Stormer             | Yes               | Yes          | 0.26         |
> > > | ArchesWeather-Mx4   | Yes               | No           | 0.74         |
> > > | **MOSAIC**              | No | No           | **1.00**         |
> > >
> > > MOSAIC achieves an energy ratio of 1.00 against ERA5 across all resolved wavelengths, confirming that our approach preserves spectral fidelity at the resolution it operates on.
> > >
> > > ### Local mass conservation and geostrophic balance near steep topography
> > >
> > > **Local mass conservation**: to address the reviewer's concern, we now provide spatially-resolved GMSP drift stratified by region and elevation (full 2020 test set, 712 init dates, 48 members):
> > >
> > > | Region              | Lead time | Mean drift (hPa) | Relative drift |
> > > |---------------------|-----------|-------------------|----------------|
> > > | Polar South         | 240h      | +0.048            | 0.005%         |
> > > | Midlat South        | 240h      | +0.090            | 0.009%         |
> > > | Tropics             | 240h      | -0.137            | 0.014%         |
> > > | **High topo (≥1500m)** | **240h** | **-0.073**     | **0.007%**     |
> > > | Low topo (<1500m)   | 240h      | -0.092            | 0.009%         |
> > > | Global              | 240h      | -0.092            | 0.009%         |
> > >
> > > All regional mean drifts remain below 0.14 hPa (0.014%) at 240h. Notably, high-topography regions show drift of 0.007%, which is comparable to the global mean and not systematically larger.
> > >
> > > **Geostrophic balance**: To assess geostrophic balance, we compute the ageostrophic fraction from geopotential and wind at 850 hPa for both a single MOSAIC ensemble member and ERA5 ground truth, averaged over 128 randomly sampled init dates in 2020:
> > >
> > > | Lead time | Region       | ERA5 ageo frac | MOSAIC ageo frac | Ratio (MOSAIC/ERA5) |
> > > |-----------|--------------|----------------|-------------------|---------------------|
> > > | 24h       | Extratropics | 0.833          | 0.830             | 1.00                |
> > > | 120h      | Extratropics | 0.833          | 0.833             | 1.00                |
> > > | 240h      | Extratropics | 0.832          | 0.831             | 1.00                |
> > > | 24h       | Midlat NH    | 1.473          | 1.394             | 0.95                |
> > > | 240h      | Midlat NH    | 1.473          | 1.380             | 0.94                |
> > > | 24h       | Midlat SH    | 0.312          | 0.319             | 1.02                |
> > > | 240h      | Midlat SH    | 0.311          | 0.321             | 1.03                |
> > >
> > > Individual MOSAIC members reproduce the ageostrophic structure of ERA5 with no degradation over 10-day rollouts as evident from ratios close to 1.0, indicating that there is no evidence of geostrophic balance breakdown.
> > >
> > > **Water budget**: We would like to emphasize that water budget analysis is *not feasible for any MLWP* (such as GraphCast, GenCast, Pangu-Weather, and Aurora), including MOSAIC, as these models do not predict precipitation, evaporation, or cloud condensate required for the analysis. We therefore, omit it in our study as well and instead provide quantitative analysis for variables that we can compute (GMSP, geostrophic balance) as well as case study of Hurricane Ian.
> > >
> > > ### BSA boundary artifacts at HEALPix base pixel boundaries
> > >
> > > We appreciate this question, and our analysis demonstrates that the answer is no, BSA does not introduce any spatial continuity artifacts. Such artifacts would reveal themselves in both spectral analysis (as they do for Stormer, which does have discontinuities due to patchification) and in rollout trajectories (which we provide in the appendix). We now include the spatial RMSE maps with boundary overlays in the revised appendix; the error fields are spatially smooth with no visible discontinuities along HEALPix edges.

---

### Official Review · Reviewer_16vH · 2026-03-13

**Soundness:** 3
**Presentation:** 2
**Significance:** 2
**Originality:** 2
**Overall Recommendation:** 3
**Confidence:** 4

**Summary:**

The paper introduces MOSAIC which is a probabilistic ML based weather forecasting model designed to address the problem of spectral degradation in weather prediction.  The idea is to combine block-sparse attention used to process high-resolution weather data without compressive encoding. This is done with the help of derived functional forecasting. The claim is that the 2 design choices help in preserving fine-scale spectral characteristics under low computational resources.  Experiments done by them on the WeatherBench style evaluation show that the model performs well, despite the constraint of operating on 1.5° resolution, and sometimes better than the finer 0.25° resolution.

**Compliance With Llm Reviewing Policy:**

Affirmed.

**Final Justification:**

Thank you to the authors for the detailed rebuttal and the additional experiments. I appreciate the effort and acknowledge that the responses address most of the questions I raised. However, I would like to flag a few specific issues.

**On the same-resolution comparison, MOSAIC is not competitive in accuracy.** The same-resolution table (240h lead time) shows:

- NeuralGCM ENS: Z500 = 606.84
- ArchesWeatherGen: Z500 = 610.36
- MOSAIC: Z500 = **624.08**

MOSAIC has a higher (worse) Z500 RMSE than both NeuralGCM and ArchesWeatherGen at the same 1.5° resolution. The authors characterise this as "competitive," but a gap of ~14-17 units on Z500 at 240h is meaningful. The speed advantage (11.6s vs 236.6s for ArchesWeatherGen) is real and impressive, but it does not resolve the accuracy gap. The paper's framing should more honestly reflect this trade-off rather than presenting MOSAIC as matching state-of-the-art accuracy.

**On the ablations, reduced-scale training may not be representative.** The 8 ablation experiments were conducted at 100k steps, batch size 1, on 4×A6000 GPUs, which is substantially below the full training scale. It is well known that relative contributions of architectural components can shift significantly between small-scale and full-scale regimes. I would like to understand whether the authors have any evidence that these ablation conclusions hold at full training scale.

**On 0.25° scaling, the evidence is indirect.** The authors trained a 1.0° model and provide a runtime extrapolation for 0.25°. This is useful context, but it does not demonstrate that the architecture remains stable and efficient at 0.25° in practice. The 1.0° checkpoint is a positive step and I welcome its release, but the scaling question raised in my original review — which I explicitly flagged as important — remains empirically unvalidated at the target resolution.

**On extreme event evaluation, one case study is insufficient.** The Hurricane Ian case study is interesting and well-presented, but a single event is not systematic evidence that spectral fidelity translates to improved extreme event skill. A comparison across multiple events or a skill score on a curated extreme event benchmark would be more convincing.

I recognise the authors committed significant effort to this rebuttal and I acknowledge that the direction of the work is promising. My original review stated I would raise my rating if the ablations and additional results were addressed. I want to be transparent that the accuracy gap in the same-resolution table and the reduced-scale ablation concern are the specific reasons I am not yet ready to do so.

**Key Questions For Authors:**

I have a few questions:

1) Can the authors provide stronger ablations to separate the contribution of block-sparse attention from the contribution of probabilistic training?

2) Since the main baselines are trained at finer resolution, how can we interpret the fairness of the comparison?

3) The paper emphasizes spectral fidelity, but do the gains in the spectral alignment translate clearly into improved forecasts for specific or small-scale events?

4) The paper mentions future scaling to 0.25° resolution, but do the authors have some evidence that the proposed design will remains stable and efficient at that resolution?

I like the paper and the presentation. If you can address the additional results and the ablations I shall raise my rating accordingly. Look forward to the rebuttal period.

**Limitations:**

Yes

**Strengths And Weaknesses:**

**Strengths**

1) The paper attempts to study an important problem in ML-based weather forecasting: preserving realistic spectral behaviour compared to optimizing the forecast metrics. The motivation is clearly explained.

2) The combination of block-sparse attention and probabilistic forecasting is interesting and the block-level design looks to be appropriate for large spatial grids and the use of HEALPix plus hardware-aligned sparse attention seems to be a valid contribution.

3) The empirical results also look promising and the paper report competitive RMSE and CRPS results against strong baselines and also shows promise for improved fidelity and reasonable ensemble calibration. The claim that MOSAIC reaches competitive performance with lower training compute is notable.

4) The paper is well-structured and the problem setup, method and experimental questions are easy to follow.



**Weakness**

1) The paper's importance is strongest at the proof-of-concept level. the results are encouraging, but broader validation across additional resolution, and operational settings are needed to claim that this is a major advance.

2) The main limitation is that the model operates at 1.5° resolution, while many of the strongest baselines are trained and evaluated on finer 0.25° data. The authors make a good case for efficiency but it seems we are making indirect comparisons as the paper itself notes that it cannot show the method behaves at the 0.25° resolution.

3) The paper also combines several ingredients at once : HEALPix interpolation, block sparse attention, U-Net style hierarchy and learned perturbations. The overall design makes sense, but would need stronger ablations to learn how every isolated component contributes to the final gains.

---

> ### Author Rebuttal · Authors · 2026-03-28
>
> We thank the reviewer for the encouraging assessment and appreciate the actionable feedback. We hope to address all four questions with additional experiments we conducted.
>
> ### Ablations separating BSA from probabilistic training
>
> We ran 8 ablation experiments, each modifying exactly one aspect relative to a controlled baseline (ERA5 2007–2018, 100k steps, batch 1, 4×A6000, no multi-step finetuning). Most relevant to this question:
>
> 1) Probabilistic vs deterministic training:
>
> | Ablation | Variant | nRMSE |
> |----------|---------|-------|
> | Training objective | Deterministic (MSE, no noise) | 0.1721 |
> | | Probabilistic (CRPS; baseline) | 0.1705 |
>
> 2) The role of BSA branches:
>
> | Ablation | Variant | nRMSE |
> |----------|---------|-------|
> | BSA sparsity | Local-only (no sparse blocks) | 0.1724 |
> | | Halved sparse blocks (12/6/2) | 0.1708 |
> | | Baseline (24/12/4) | 0.1705 |
> | | Doubled sparse blocks (48/24/8) | 0.1702 |
> | Selection branch | Local + compression only (no selection) | 0.1720 |
> | | Full BSA (baseline) | 0.1705 |
>
> From those studies, we conclude that both BSA's sparse selection and probabilistic training contribute independently.
>
> **Action taken**: Extensive ablation study (8 experiments) added to the appendix.
>
> ### Fairness of comparison
>
> We now provide a same-resolution benchmark against models operating at 1.5°, all trained on ERA5 up to 2019, tested on 2020. Results at 240h lead time:
>
> | Model | Resolution | Ensemble time*, s | Z500 | T850 | Q700 | U850 |
> |-------|-----------|------|------|------|------|------|
> | NeuralGCM ENS (50) | 1.4° | 524 | **606.84** | 2.756 | 1374.2 | 4.830 |
> | ArchesWeatherGen | 1.5° | 236.6 | 610.36 | **2.755** | 1373.7 | 4.830 |
> | **MOSAIC (Ours)** | **1.5°** | **11.6** | 624.08 | 2.778 | **1358.1** | 4.880 |
> | ArchesWeather-Mx4 | 1.5° | — | 693.56 | 3.117 | 1541.1 | 5.413 |
> | Stormer ENS | 1.4° | — | 665.88 | 3.001 | 1445.4 | 5.198 |
>
> \* 24-member, 10-day rollout, single H100.
>
> MOSAIC is competitive with ArchesWeatherGen and NeuralGCM while being 20–45× faster for ensemble generation, enabling real-time probabilistic forecasting.
>
> **Action taken**: Controlled same-resolution benchmark added against state-of-the-art 1.5° models.
>
> ### Spectral fidelity translating to improved forecasts
>
> We address this on three levels. First, spectral comparison at 1.5° against ArchesWeatherGen, Stormer, and ERA5 confirms that spectral degradation occurs even at coarse resolution (Stormer shows clear suppression), while MOSAIC preserves the spectrum up to the Nyquist limit.
>
>  Second, we generated 10-day forecast rollouts for wind and temperature fields which we provide in appendix. MOSAIC maintains sharp, physically plausible features throughout the forecast horizon without the progressive blurring seen in deterministic baselines.
>
> Third, we added a Hurricane Ian case study: 48-member ensemble initialized Sep 23 2022, 5 days before Category 4 landfall. We use Extreme Weather Bench methodology and demonstrate that MOSAIC successfully captures the hurricate at 5 day lead time within the enseble and correctly simulates the recurvature over Cuba and subsequent landfall on the Florida Gulf Coast.
>
> Fourth, we evaluated model stability via a global mass conservation diagnostic (GMSP drift over ten-day rollouts in test year). The maximum mean drift is 0.086 hPa (0.009% relative to ~1013 hPa), so the model neither creates nor destroys atmospheric mass over extended rollouts, suggesting stable dynamics.
>
> **Action taken**: We conduct a Hurricane Ian case study with track verification and wind field evolution, and mass conservation study over test year.
>
> ### Evidence for 0.25° scaling
>
> We trained a MOSAIC model on 1.0° data (~65k tokens) and did not observe any difficulties during training -- we will release the checkpoint upon acceptance.
>
> That being said, we do lack the computational resources for 0.25° training (780k+ tokens per sample, 16× more than 1.5°). However, the architecture itself has no resolution-dependent barriers. Functional perturbations are validated at 0.25° by Alet et al.in FGN, and BSA remains tractable at scale. Notably, we conducted a runtime scaling study and found that BSA processes 780k tokens in 320 ms vs *32,000 ms* for dense FlashAttention (100× speedup):
>
> | Sequence length | BSA (ms) | NSA (ms) | FlashAttention (ms) | BSA / FlashAttention |
> |----------------|----------|----------|---------------------|----------------------|
> | 8k | 3.0 | 7.5 | 2.7 | 0.9× |
> | 16k | 6.3 | 17.9 | 11.2 | 1.8× |
> | 32k | 13.1 | 36.5 | 44.8 | 3.4× |
> | 64k | 27.8 | 86.0 | 176.9 | 6.4×*|
> | 128k | 57.2 | 234.1 | 752.7 | 13.2× |
> | 256k | 115 | 700 | 3,384 | 29.5× |
> | 512k | 231 | 2,115 | 14,259 | 61.8× |
> | 780k (0.25° resolution) | 327 | 3,169 | 32,139 | 98.3× |
>
> **Action taken**: We add runtime scaling anaylysis against both NSA and Dense FlashAttention at scale of 0.25° in the appendix. We also now provide a 1° model checkpoints.

---

> > ### Author Rebuttal · Reviewer_16vH · 2026-04-04
> >
> > Thank you to the authors for the helpful rebuttal. The responses have addressed most of my concerns satisfactorily, and I do not have any further questions at this stage. I would like to stay with my current rating of the paper.

---

> > > ### Author Response · Authors · 2026-04-04
> > >
> > > We thank the reviewer for engaging with our rebuttal and for confirming that our responses addressed the raised concerns satisfactorily. We are glad to hear there are no remaining questions.
> > >
> > > We would like to respectfully note that the original review stated: "If you can address the additional results and the ablations I shall raise my rating accordingly." In our rebuttal, we provided (1) 8 ablation experiments isolating each component's contribution, (2) a controlled same-resolution benchmark against state-of-the-art 1.5° models, (3) a Hurricane Ian case study demonstrating spectral fidelity translating to improved extreme event forecasting, and (4) runtime scaling evidence to 0.25° resolution. The reviewer has confirmed these responses are satisfactory with no outstanding issues.
> > >
> > > Given that the specific conditions for raising the score appear to have been met, we would welcome clarification on what additional concerns motivate the current rating, so that we may address them.

---

### Official Review · Reviewer_skwX · 2026-03-13

**Soundness:** 3
**Presentation:** 3
**Significance:** 2
**Originality:** 3
**Overall Recommendation:** 5
**Confidence:** 3

**Summary:**

This paper introduces MOSAIC, a probabilistic Large Weather Model designed to mitigate spectral degradation in ML-based weather forecasting. The authors propose integrating functional perturbations (noise injected into SwiGLU gates) for ensemble generation and a Block Sparse Attention mechanism operating on a contiguous HEALPix mesh. The authors claim that by avoiding compressive encoding, MOSAIC preserves fine-scale spectral fidelity and matches the performance of 0.25-degree state-of-the-art models despite operating at a significantly coarser 1.5-degree resolution with a fraction of the compute.

**Compliance With Llm Reviewing Policy:**

Affirmed.

**Final Justification:**

I am raising my score to an Accept. The authors provided a detailed and transparent rebuttal that adequately addressed my primary concerns. The paper presents a practical system-level integration by adapting Native Sparse Attention to the HEALPix mesh alongside functional perturbations. The supplementary experiments helped validate the soundness of their empirical claims. Furthermore, their willingness to revise absolute claims and discuss the architectural limitations regarding spatio-temporal modeling improves the clarity and rigor of the manuscript.

**Key Questions For Authors:**

Please see the weaknesses.

**Limitations:**

yes

**Strengths And Weaknesses:**

Strengths:
1. The hardware-aware engineering of the Block Sparse Attention module is commendable. Adapting Native Sparse Attention (NSA) to operate at a block-to-block level on the HEALPix mesh to ensure contiguous memory access and coalesced GPU reads is a highly practical system optimization.

2. The approach to uncertainty quantification is computationally elegant. Injecting noise directly into the SwiGLU activations to produce adaptive, input-dependent covariance for the ensemble members is a lightweight alternative to running multiple independent diffusion models.

Weaknesses:
1. My primary and most critical concern lies in a logical disconnect regarding the spectral fidelity claims and the model's operating resolution. The authors heavily criticize baselines for losing spectral power at fine scales. Fig. 2 shows degradation below 500km wavelengths for 0.25-degree models. However, MOSAIC operates at a 1.5-degree resolution, which inherently has a Nyquist limit of roughly ~333km (as seen in Fig. 6). Claiming that MOSAIC preserves spectral fidelity is misleading when the model fundamentally cannot resolve the mesoscale frequencies (50km-300km) where state-of-the-art models actually struggle. Preserving energy at coarse synoptic scales is not equivalent to solving the spectral degradation problem.

2. While the authors avoid Vision Transformer-style patchification, Section 4.3 explicitly states that the U-Net encoder "progressively coarsens the HEALPix mesh" aggregating 4 children pixels into 1 parent pixel. This hierarchical spatial pooling is a form of compressive encoding and information bottlenecking.

3. Comparing a native 1.5-degree model against 0.25-degree models regridded to 1.5 degrees masks the well-known "double penalty" effect in meteorology, where high-resolution models are heavily penalized in RMSE for minor spatial displacements of fine-scale features that coarse models simply smear out.

---

> ### Author Rebuttal · Authors · 2026-03-28
>
> We thank the reviewer for recognizing the engineering contributions of BSA and the elegance of functional perturbations, and address the remaining concerns below.
>
> ### Spectral fidelity claims at 1.5° resolution
>
> We admit that our original framing was insufficiently precise, and we have revised the manuscript accordingly. The claim we make is that MOSAIC *does not* introduce any systematic spectral degradation at the frequencies the grid can resolve. Basically, it learns the weather evolution function as faithfully as the data resolution permits, and that is the main point of the paper. This matters because spectral degradation is not specific to fine grid: it occurs at any resolution when the architecture or training procedure fails to preserve the information the grid can represent.
>
> To illustrate the point, we add a comparison against other state-of-the-art models that were natively trained on 1.5° data, both probabilistic (ArchesWeatherGen) and deterministic (ArchesWeather-Mx4, and Stormer). We report the ratio of predicted to ERA5 kinetic energy at resolved wavelengths for 10m wind speed (10-day forecast, 1.5°):
>
> | Model | Energy ratio |
> | :--- | :--- |
> | **MOSAIC (single member, Ours)** | **1.001** |
> | MOSAIC (ensemble mean, Ours) | 0.297 |
> | ArchesWeatherGen (single member) | 1.025 |
> | ArchesWeatherGen (ensemble mean) | 0.233 |
> | ArchesWeather-Mx4 | 0.742 |
> | Stormer | 0.256 |
>
> What we demonstrate then is that the pattern we see for high-resolution models holds just as well for coarse models: deterministic models show clear spectral suppression at wavelengths the grid resolves, while MOSAIC and ArchesWeatherGen closely track ERA5's spectrum up to the Nyquist limit.
>
> **Action taken**: We added an apple-to-apple spectral comparison against other 1.5° models where we demonstrate that deterministic models induce spectral degradation.
>
> ### U-Net coarsening as compression
>
> This is a fair observation, however, the key distinction here is not whether coarsening happens, but when. Specifically, MOSAIC natively operates at full resolution and computes spatial interactions over fine details *before* coarsening, with skip connections carrying the fine-scale information all the way through the model. This is different from compression-based models, where spatial interactions are captured on the latent (compressed) grid -- tokens never interact at native resolution.
>
> We empirically validate the distinction in one of our ablations: we skip the native-resolution stage and instead use cross-attention to map to coarse grid directly (keeping the number of stages / hidden dimensions / parameters comparable):
>
> | Model | nRMSE | Energy ratio |
> | :--- | :--- | :--- |
> | **MOSAIC (native res.)** | **0.1705** | **1.001** |
> | MOSAIC (compress-first) | 0.1771 | 0.974 |
>
> This is a very considerable degradation. For comparison, reducing the number of parameters from 220M to 110M yields nRMSE of 0.1746. This indicates that added capacity cannot compensate for compression-induced information loss, which is consistent with findings in computer vision literature (Wang et al., 2025) .
>
> **Action taken**: We provide a spatial compression ablation and clarify the distinction from compress-first architectures.
>
> ### Double-penalty effect in RMSE comparisons
>
> This is a valid concern, which we mitigate through using the established evaluation protocol: all models are evaluated under WeatherBench 2 (Rasp et al., 2023) with conservative regridding to 1.5°, which is the community standard. We also measure CRPS which directly penalizes ensemble miscoverage. MOSAIC demonstrates competitive performance: at 240h lead time, Z500 CRPS: MOSAIC 258 vs IFS-ENS 261 vs GenCast 254. Furthermore, the ordering under CRPS is consistent with RMSE ordering, suggesting the double-penalty effect is not artificially inflating MOSAIC's relative performance.
>
> Additionally, we now provide a comparison of MOSAIC against state-of-the-art 1.5° models (all models trained on ERA5 data up to 2019, and tested on 2020 year), where we demonstrate that MOSAIC is on par in terms of performance but is an order of magnitude faster than of its probabilistic counterparts, enabling real-time probabilistic forecasting. Below are the results at 240h lead time:
>
> | Model | Resolution | Inference time **, s | Z500 (m²/s²) | T850 (K) | Q700 (mg/kg) | U850 (m/s) |
> |-------|-----------|------|------|------|------|------|
> | NeuralGCM ENS (50) | 1.4° | 524 | **606.84** | 2.756 | 1374.2 | 4.830 |
> | ArchesWeatherGen (mean) | 1.5° | 236.6 | 610.36 | **2.755** | 1373.7 | 4.830 |
> | **MOSAIC (mean; Ours)** | 1.5° | **11.6** | 624.08 | 2.778 | **1358.1** | 4.880 |
> | ArchesWeather-Mx4 | 1.5° | -- | 693.56 | 3.117 | 1541.1 | 5.413 |
> | Stormer ENS (mean) | 1.4° | -- | 665.88 | 3.001 | 1445.4 | 5.198 |
>
> ** Inference time is reported for a 10 day rollout of 24-member ensemble on a single H100 GPU.
>
> **Action taken**: We provide a comparison against other 1.5° models.

---

> > ### Author Rebuttal · Reviewer_skwX · 2026-04-03
> >
> > Thank you to the authors for the detailed and high-quality rebuttal. The newly added same-resolution (1.5 degree) baseline comparison experiments are highly convincing. They effectively alleviate my concerns regarding evaluation fairness and the double-penalty effect, successfully demonstrating MOSAIC's true competitiveness at this scale. Adapting Native Sparse Attention (NSA) to the HEALPix spherical mesh and combining it with parameter noise injection for ensemble generation is an interesting and promising system-level integration. After carefully reading your response, I agree that this work makes clear progress in computational efficiency for large spatial grids. However, to further strengthen the rigor of the paper, there are still a few potential caveats and methodological boundaries that I hope we can discuss further in this ongoing review phase.
> >
> > First, I would like to explore the potential accumulation of interpolation errors during autoregressive generation. The model employs a cross-attention mechanism to transition between the latitude-longitude grid and the HEALPix mesh. This means that during a 10-day continuous rollout, this resampling occurs frequently at every time step. Given that cross-attention inherently introduces some spatial smoothing effects, I am curious whether this repeated interpolation might accumulate implicit errors over multiple iterations, potentially conflicting with the core claim of "preserving high-frequency spectral details." Furthermore, it would be helpful to clarify the actual impact of this conversion overhead on the overall inference efficiency. I would appreciate it if the authors could share their insights on the trade-offs between continuous latent space rollouts and the errors introduced by explicit step-by-step reconstruction.
> >
> > Second, I would like to further discuss the continuity of spatio-temporal modeling. The current Block Sparse Attention (BSA) mechanism appears to be designed primarily for single-frame spatial snapshots. If the model merely stacks historical time steps as static feature channels without introducing explicit spatio-temporal joint attention in the core layers, could this lead to temporal inconsistencies or prediction lag when tracking rapidly moving weather systems? Given the strong spatio-temporal coupling in fluid dynamics, I wonder where the capability boundary lies for a purely spatial attention architecture in capturing highly dynamic features.
> >
> > Finally, regarding the rationale of using noise injection to generate ensemble members. Utilizing Functional Perturbations (SwiGLU noise injection) indeed improves the statistical performance of ensemble spread and yields better CRPS scores. However, from a generative modeling perspective, how can we ensure that this parameter-level random perturbation is truly learning meteorologically meaningful "uncertainty structures"? Does the resulting local high-frequency fluctuation reflect genuine extreme weather variability, or is it merely a random artifact generated to satisfy statistical variance? Relying solely on a single hurricane case study or the macroscopic statistics of Global Mean Surface Pressure (GMSP) might not fully substantiate the physical plausibility of this purely statistical perturbation under complex extreme weather conditions.
> >
> > Overall, this is a promising and valuable work that makes clear exploratory steps in architectural optimizations for meteorological grid computation. If the authors could provide further objective discussion or qualitative analysis regarding the aforementioned questions on accumulated errors, spatio-temporal limitations, and the rationale behind the perturbations, while perhaps slightly softening absolute claims such as operating "completely without compression," it would make the paper's conclusions much more robust. I look forward to your thoughts on these points.

---

> > > ### Author Response · Authors · 2026-04-04
> > >
> > > We thank the reviewer for these thoughtful follow-up questions and are happy to engage in a discussion hoping to improve the paper.
> > >
> > > ### Interpolation error accumulation during autoregressive rollouts
> > >
> > > Error accumulation in the spectral domain was the single most important metric for us to track during model development. For cross-attention specifically, we did not observe any negative effects. We attribute this to the fact that interpolation happens in the latent space rather than in the physical space, as input features are projected into a latent representation before cross-attention.
> > >
> > > Early in the project, we tried bilinear interpolation to the HEALPix grid followed by model application and interpolation back. This produced strong visible spatial smoothing, motivating further search as we had to operate on HEALPix for the contiguous memory layout but without sacrificing fidelity.
> > >
> > > Cross-attention was the solution we converged on after experimenting with MPNNs, and we did not observe it inducing negative effects during unrolling. MOSAIC's RMSE growth over lead time is smooth and consistent with other state-of-the-art models. Moreover, at 10-day lead time the single-member energy ratio is 1.001, essentially perfect spectral preservation across all resolved wavelengths.
> > >
> > > Regarding inference efficiency, interpolation accounts for less than 5% of overall runtime. It essentially reduces to a series of matmuls with softmax applied only over the small neighbor list (24 in our case), which is extremely cheap.
> > >
> > > **On latent rollouts.** To our knowledge, this is an active area of research. We would be very interested to see its effect on spectral fidelity, but our intuition is that it would be less capable of faithfully capturing fine-scale dynamics. This remains an open question.
> > >
> > > ### Spatio-temporal modeling in BSA
> > >
> > > MOSAIC processes temporal information by stacking historical time steps as input feature channels. We agree this is a design boundary worth discussing openly.
> > >
> > > This temporal encoding strategy is standard across medium-range MLWPs: GraphCast, Aurora, and GenCast all follow the scheme. The temporal modeling mechanism then boils down to the autoregressive rollout, which empirically is sufficient for modelling the evolution of weather systems (e.g. the Hurricane Ian study) with 6h / 24h step length.
> > >
> > > That being said, for very rapid phenomena with fine temporal discretization (e.g. nowcasting), explicit spatio-temporal attention could offer benefits as it would allow to process large data (hence more information) in a scalable way. This is an interesting direction for future work, particularly as models scale to finer resolutions where such dynamics become resolvable. We will add this discussion to the limitations section.
> > >
> > > ### Physical plausibility of functional perturbations
> > >
> > > The key design rationale, following FGN (Price et al., 2025), is that modeling uncertainty in parameter space produces more structured variations than perturbing inputs or outputs directly. All stochasticity originates from a single low-dimensional noise vector shared across all spatial locations. Because these parameters are reused across space, the network learns to transform the perturbation into spatially coherent atmospheric states.
> > >
> > > We completely agree that physical consistency is important and provide two additional pieces of evidence:
> > >
> > > **Ageostrophic fraction at 850 hPa.** We compute the ratio of ageostrophic to total wind speed for a single MOSAIC member vs ERA5. The MOSAIC/ERA5 ratio is 1.00 across the extratropics at all lead times up to 240h, with per-region ratios of 0.94–1.03 (Midlat NH: 0.94, Midlat SH: 1.03 at 240h). Individual members preserve the ageostrophic wind structure throughout the rollout with no degradation.
> > >
> > > **Spatially pooled CRPS.** Following FGN's protocol, we compute CRPS after average- and max-pooling within spatial windows of increasing size. If perturbations were per-grid-point noise, spread would cancel under averaging. Z500 results (T2m, MSLP, U10 show the same pattern):
> > >
> > > | Pool size | Avg-pool CRPS (24h / 120h / 240h) | Max-pool CRPS (24h / 120h / 240h) |
> > > |-|-|-|
> > > | per-point | 23.3 / 116.0 / 264.4 | 23.3 / 116.0 / 264.4 |
> > > | 480 km    | 21.3 / 112.2 / 259.5 | 21.8 / 106.6 / 247.2 |
> > > | 1920 km   | 15.5 / 83.4 / 210.3  | 18.5 / 76.9 / 181.1  |
> > > | 3828 km   | 11.9 / 57.0 / 146.2  | 16.9 / 61.3 / 135.9  |
> > >
> > > Both metrics decrease monotonically with pool size, confirming spatially coherent spread and realistic regional extremes.
> > >
> > > ### On the "without compression" claim
> > >
> > > We appreciate this suggestion. In the revised manuscript, we no longer claim that no spatial reduction occurs anywhere in the pipeline -- rather, that MOSAIC computes spatial interactions at native resolution *before* any coarsening occurs thus avoiding the information bottleneck of compress-first architectures where tokens never interact at native resolution.

---

### Decision · Program_Chairs · 2026-04-30

**Decision:**

Accept (regular)

**Comment:**

The paper received four reviews with scores of 5, 3, 3, 5. Post-rebuttal, skwX raised to Accept and XhZa raised to 5; 16vH and xWoH did not update despite receiving substantive responses. The rebuttal was particularly strong, delivering same-resolution benchmarks, 8 ablations, optimizer comparisons, runtime scaling projections, etc... Even though two reviewers are still leaning on the reject side, their comments can be interpreted as a borderline assessment.

Overall, the area chair believes the paper will be a nice contribution for ICML, but all the material introduced during the discussion should be included in the camera ready version of the paper.